# ENTROPY-REGULARIZED MODEL-BASED OFFLINE REINFORCEMENT LEARNING

## ABSTRACT

Model-based approaches to offline Reinforcement Learning (RL) aim to remedy the problem of sample complexity in offline learning via first estimating a pessimistic Markov Decision Process (MDP) from offline data, followed by freely exploring in the learned MDP for policy optimization. Recent advances in model-based RL techniques mainly rely on an ensemble of models to quantify the uncertainty of the empirical MDP which is leveraged to penalize out-of-distribution state-action pairs during the policy learning. However, the performance of ensembles for uncertainty quantification highly depends on how they are implemented in practice, which can be a limiting factor. In this paper, we propose a systematic way to measure the epistemic uncertainty and present EMO, an Entropy-regularized Model-based Offline RL approach, to provide a smooth error estimation when leaving the support of data toward uncertain areas. Subsequently, we optimize a single neural architecture that maximizes the likelihood of offline data distribution while regularizing the transitions that are outside of the data support. Empirical results demonstrate that our framework achieves competitive performance compared to state-of-the-art offline RL methods on D4RL benchmark datasets.

## 1 INTRODUCTION

Following the major success of deep Reinforcement Learning (RL) in numerous applications (Mnih et al., 2013; 2015; Silver et al., 2018), offline RL has emerged to cope with the problems where simulation or online interaction is impractical, costly, and/or dangerous, thus, allowing to automate a wide range of decision-making problems from healthcare and education to finance and robotics (Levine et al., 2020). The primary challenge in these scenarios is however that learning new policies from data stored with a different (possibly sup-optimal) policy, aka behavior policy, suffers from distributional shifts resulting in *extrapolation error*, which is infeasible to improve due to lack of additional exploration (Fujimoto et al., 2019; Kumar et al., 2019). This is why standard (online) RL methods perform poorly in offline settings (Yu et al., 2020). Consequently, several model-free offline RL algorithms are introduced to regularize the learned policies to stay close to the behavior policy, by constraining out-of-distribution trajectories (Fujimoto et al., 2019; Kumar et al., 2019; Wu et al., 2019; Kumar et al., 2020; Agarwal et al., 2020).

In model-free methods, policy optimization is limited to already observed states which most likely do not provide sufficient coverage of the entire state space. Alternatively, model-based methods first learn the corresponding empirical Markov Decision Process (MDP) using the offline dataset and then freely explore in the learned environment for policy optimization, which can attain excellent sample efficiency compared to model-free methods (Chua et al., 2018; Janner et al., 2019). Most recently, model-based algorithms are specifically designed for offline settings to address distributional shifts in the learned model and have been proved effective in certain problems compared to their model-free counterparts (Yu et al., 2020; 2021; Kidambi et al., 2020; Zhan et al., 2021; Swazinna et al., 2021; Chen et al., 2021; Rigter et al., 2022)

However, prominent model-based methods, i.e., MOPO (Yu et al., 2020) and MOReL (Kidambi et al., 2020), mainly leverage an ensemble of models for uncertainty quantification. Ensemble uncertainty quantification is a special case of uncertainty quantification in Bayesian neural networks with latent variables using nearest-neighbor methods, introduced by Depeweg et al. (2018), where each model in the ensemble corresponds to sample from the posterior distribution. In these methods,

a measure of ensemble discrepancy determines the estimation error. This can be particularly restrictive when theoretical assumptions on the ensemble do not hold in practical scenarios. In practice, an ensemble usually consists of a small number of models, where each model is a different initialization of the same neural architecture, trained on the same data. Hence, the models in the ensemble are likely to correlate to one another after training, which might make their variation a poor indicator of uncertainty. Yu et al. (2021) study this behavior and demonstrate that the uncertainty estimated via maximum variance over the learned ensemble (as in MOPO) struggles to accurately predict the model's error , and could lead to poor performance (see Fig. 2 in Yu et al. (2021)).

Accordingly, there have been efforts to eliminate the need for bootstrap ensembles for uncertainty estimation in model-based offline RL methods. Yu et al. (2021) propose utilizing model rollouts to conservatively learn the Q-function by penalizing the values over out-of-distribution areas, while Rigter et al. (2022) introduce an adversarial framework for training the policy and the model at the same time, such that at each step, the policy is trained to maximize the return, while the model is tuned to minimize it. In addition, Tennenholtz et al. (2021) propose to quantify uncertainty using a $k$-nearest neighbors approach, where the distance measure is defined as an approximate metric on the learned (Riemannian) manifold in a latent space encoded by a VAE. Although RAMBO-RL (Rigter et al., 2022) and COMBO (Yu et al., 2021) have shown promising empirical results on standard benchmark datasets, they both forgo the modularity aspect of methods such as MOPO and MOReL, and GELATO (Tennenholtz et al., 2021) is computationally expensive. Instead, we aim to get the best of both worlds and present a general-purpose, task-agnostic, computationally efficient framework to learn a pessimistic model of the environment that can be coupled with any RL algorithm to learn optimal policies, without ensemble learning.

In this paper, we address this problem by proposing a novel method that eliminates the need for ensemble uncertainty quantification, while still being modular in the sense that the trained model can be combined with arbitrary RL algorithms to learn arbitrary tasks. Therefore, we present EMO, an Entropy-regularized Model-based Offline RL approach, that learns a pessimistic MDP using only a single model which can provide accurate estimates of the dynamics in the support of offline data while softly quantifies an upper bound for the uncertainty of model predictions when leaving the data support. To this end, we devise a regularized loss function to minimize the negative log-likelihood of the model w.r.t. the offline data distribution, and simultaneously, maximize the entropy of predictions outside of the data support in a single model. Furthermore, we propose to warm-start the learning procedure by only optimizing the unconstrained objective function, where the initial learned model in this step is used to generate rollouts for optimizing the uncertainty estimation.

Our extensive empirical study illustrates that our approach achieves better or on par performance compared to state-of-the-art (SOTA) offline RL techniques, both model-free and model-based, on D4RL benchmark datasets for MuJoCo environments.

## 2    RELATED WORK

Offline reinforcement learning (Lange et al., 2012), which allows for optimizing policies from static offline datasets, has received a lot of attention throughout the recent years, as the practical issues of applying online RL to many real-world scenarios became more apparent. Model-free offline RL approaches optimize a policy solely based on the visited states from the static offline data, without utilizing a learned model of the environment. Constraining the policy to be close to the behavior policy (Kumar et al., 2019; Fujimoto et al., 2019; Wu et al., 2019; Fujimoto & Gu, 2021), conservative estimation of value functions (Kumar et al., 2020; Kostrikov et al., 2021), incorporating the uncertainty of predictions to stabilize Q-functions (Agarwal et al., 2020; Wu et al., 2021), and adversarial training of actor and critic (Cheng et al., 2022) are among active lines of work in model-free offline RL. However, due to their limited generalization, the performance of model-free methods is highly reliant on the optimality of the offline data.

On the other hand, model-based approaches incorporate a model of the environment to improve generalization and sample efficiency, which is used as a surrogate for the actual MDP to optimize a policy, combined with the original offline data. MOPO (Yu et al., 2020) and MOReL (Kidambi et al., 2020) incorporate ensemble uncertainty estimation to penalize highly uncertain transitions. COMBO (Yu et al., 2021) combines the idea of conservative estimation of value functions in CQL (Kumar et al., 2020) with a model-based learning framework. RAMBO-RL (Rigter et al., 2022)

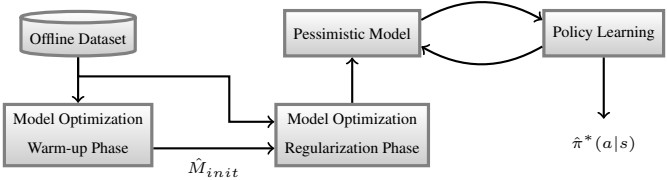

Figure 1: General scheme of EMO.

proposes an adversarial framework for training the policy and the model at the same time, such that at each step, the policy is trained to maximize the return, while the model is tuned to minimize it.

As discussed in (Yu et al., 2021), algorithms like MOPO and MOReL that rely on ensemble uncertainty estimation, are prone to erroneous estimations of uncertainty. On the other hand, COMBO and RAMBO-RL are not modular frameworks; COMBO applies the idea of conservative learning of value function as a part of its proposed RL algorithm, and RAMBO-RL tunes the model to be pessimistic with respect to the current policy of the agent; therefore, the resulting model from learning a particular task with either of these methods cannot be utilized to learn a new task in the environment. EMO, however, tries to eliminate the need for ensemble uncertainty quantification, while remaining a modular framework that can be extended to various tasks and RL algorithms.

## 3 EMO: ENTROPY-REGULARIZED MODEL-BASED OFFLINE RL

### 3.1 PRELIMINARIES

The Reinforcement Learning (RL) problem is characterized via a Markov Decision Process (MDP) (Sutton & Barto, 2018), represented by $M = (\mathcal{S}, \mathcal{A}, T, r, \gamma, \rho_0)$, where $\mathcal{S}$ and $\mathcal{A}$ denote the state space and action space, respectively, $T(s'|s, a)$ is the transition distribution, $r(s, a)$ is the reward function, $\gamma \in (0, 1)$ is the discount factor, and $\rho_0$ is the distribution of the initial state. A policy $\pi(a|s)$ is defined as a mapping from states to a distribution over actions, $\pi : \mathcal{S} \times \mathcal{A} \to [0, 1]$, and the goal is to learn a policy $\pi^*$ which maximizes the expected discounted return $\eta_M(\pi)$ when followed

$$\pi^* = \arg\max_\pi \eta_M(\pi) \quad \text{where} \quad \eta_M(\pi) = \mathbb{E}_{\pi, \rho_0, T}\Big[\sum_{t=0}^{\infty} \gamma^t r(s_t, a_t)\Big].$$

However, in offline settings, we are not able to accurately evaluate the return values under different policies as there is no further interaction with the environment. Instead, a static dataset $\mathcal{D} = \{(s_i, a_i, r_i, s'_i)\}_{i=1}^{n}$ which is collected under an arbitrary policy in the environment is provided. Therefore, the objective turns into finding the best policy that can be solely optimized on the available offline data $\mathcal{D}$. Note that in this context, best policy might be different from the optimal policy, as the performance of the resulting policy, regardless of the learning algorithm, is affected and capped by factors such as distribution and optimality of the static data.

In this paper, we focus on model-based offline reinforcement learning, where an empirical model of the environment is estimated and leveraged to enhance sample efficiency over model-free approaches. In this framework, the offline dataset is used to train a *pessimistic* MDP $\hat{M}$ which is then employed as a surrogate for the actual model of the environment. Subsequently, the RL agent interacts with this model and optimizes its policy based on both the acquired information and the original transitions from the offline data. Ideally, we aim to find a policy $\hat{\pi}^*$ with minimum sub-optimality with respect to the optimal policy, i.e., $\hat{\pi}^* = \arg\min_\pi \eta_M(\pi^*) - \eta_M(\pi)$.

$\hat{\pi}^* = \arg\min_\pi \eta_M(\pi^*) - \eta_{\hat{M}}(\pi)$.

The general workflow of EMO is depicted in Figure 1. In this figure, the offline dataset is first utilized to train a pessimistic model of the environment in a two-phase training regime. After that, the model is considered as a surrogate for the actual environment to train a policy $\hat{\pi}^*(a|s)$.

### 3.2 MODEL OPTIMIZATION

In this section, we present a novel approach for model optimization, as the main part of EMO, that learns a pessimistic MDP using only a single neural network with two aims. First, the estimated model should accurately capture the dynamics of the environment (which can only be reliable in the support of offline data,) and second, it should relatively quantify the uncertainty associated with model predictions in the form an error estimator. Consequently, we address these goals in a regularized optimization setting, where both the likelihood of the data as well as the error estimation are jointly optimized in a single model.

The error estimator, denoted by $u$, allows the framework to relatively differentiate between the reliable and unreliable predictions of the model, which accordingly, can be utilized to penalize unreliable predictions based on their estimated uncertainty. Let $r(s, a)$ and $u(s, a)$ be the associated reward function and estimated error for a particular state-action pair $(s, a)$, respectively. A pessimistic model can be determined by defining an alternative reward value in the form of $\tilde{r}(s, a) = r(s, a) - \lambda u(s, a)$, where $\lambda$ is a constant value to control the amount of penalty associated with the state-action pair. As a result, the policies will be prevented from exploiting unreliable predictions, which can translate to conservative exploration in the actual environment.

Inspired by Yu et al. (2020); Kidambi et al. (2020), we characterize the one-step model of the environment with a Gaussian distribution over the next state and reward, conditioned on the current state and action, i.e., $P(s', r|s, a) = \mathcal{N}\big(\mu_\theta(s, a), \Sigma_\phi(s, a)\big)$, where $\theta$ and $\phi$ are the weights of the corresponding neural networks. Both the mean vector $\mu_\theta(s, a)$ and covariance matrix $\Sigma_\phi(s, a)$ are of the size $d + 1$, where $d$ is the dimensionality of the state space and the covariance matrix is assumed to be diagonal. In this work, the main reason for explicitly estimating the covariance matrix lies in the fact that the entropy of a Gaussian distribution, as a quantitative measure for uncertainty, is formulated as a function of the covariance matrix

$$H = \frac{1}{2} \log \Big( \det \big( \Sigma_\phi(s, a) \big) \Big) + \frac{(d + 1)}{2} \big( 1 + \log(2\pi) \big),$$

and assuming that the covariance matrix is diagonal, we have:

$$H = \frac{1}{2} \sum_{i=1}^{d+1} \log \big( \Sigma_{\phi,i}(s, a) \big) + \frac{(d + 1)}{2} \big( 1 + \log(2\pi) \big). \tag{1}$$

Consequently, $\Sigma_\phi(s, a)$ can solely act as the uncertainty/error estimator, while $\mu_\theta(s, a)$ is trained to model the transition dynamics. In this way, both terms can be optimized in a single model to learn a pessimistic MDP that allows for utilizing the offline data in a more efficient and reliable way.

Furthermore, we propose a two-step algorithmic framework for learning the pessimistic model of the environment. During the first phase, indicated by the *warm-up* phase, the model is trained via maximum likelihood following the prior work (Yu et al., 2020; Kidambi et al., 2020), while in the second phase, which we call the *regularization* phase, the model is leveraged to generate synthetic rollouts, that are then used to maximize the entropy of model predictions over the out-of-distribution data points. Note that the warm-up phase is essential to the next phase, since the additionally generated data from the model needs to be as close to the actual dynamics of the environment as possible, particularly, in the support of offline data and possibly its generalizable neighborhood.

#### 3.2.1 WARM-UP PHASE

In this phase, we employ the Gaussian negative log-likelihood (NLL) loss to train an initial model $\hat{M}_{init}$ of the environment on the offline dataset. Let $\mathbb{B}_\mathcal{D}$ be a batch sampled from the offline data $\mathcal{D}$, the NLL objective function denoted by $\mathcal{L}_1$ can be written as in Equation 2

$$\mathcal{L}_1(\theta, \phi; \mathbb{B}_\mathcal{D}) = \frac{1}{|\mathbb{B}_\mathcal{D}|} \sum_{(s,a,r,s') \in \mathbb{B}_\mathcal{D}} \sum_{i=1}^{d+1} \frac{1}{2} \Big( \log(\Sigma_{\phi,i}(s, a)) + \frac{(\mu_{\theta,i}(s, a) - [s', r]_i)^2}{\Sigma_{\phi,i}(s, a)} \Big). \tag{2}$$

At the end of the warm-up phase, we expect $\mu_\theta(s, a)$ to provide accurate and reliable predictions for $(s', r)$ conditioned on $(s, a)$ in the support of the offline data. Additionally, depending on the generalizability of the model, the performance can be extended to a neighborhood around the offline

---

**Algorithm 1** Generating Rollouts

---

**Require:** $\mathcal{D}$, $\mu_\theta$, $\Sigma_\phi$, $\pi_e$, batch size $b$, rollout horizon $h$, penalty coefficient $\lambda$
1: Set $\bar{\mathbb{B}}_{\pi_e} \leftarrow \emptyset$
2: **for** $1, 2, ..., b$ (in parallel) **do**
3:      Set $\rho \leftarrow \emptyset$
4:      Sample state $s_1$ from $\mathcal{D}$ for the initialization of the rollout.
5:      **for** $j = 1, 2, ..., h$ **do**
6:          Sample an action $a_j \sim \pi_e(s_j)$
7:          Obtain $s_{j+1}, r_j = \mu_\theta(s_j, a_j)$
8:          (Optional) Compute $\tilde{r}_j = r_j - \lambda\sqrt{\text{tr}(\Sigma_\phi(s_j, a_j)}$
9:          Add $(s_j, a_j, \tilde{r}_j, s_{j+1})$ to $\rho$
10:     **end for**
11:     Add $\rho$ to $\bar{\mathbb{B}}_{\pi_e}$
12: **end for**
13: **return** $\bar{\mathbb{B}}_{\pi_e}$

---

data distribution. Similarly, $\Sigma_\phi(s, a)$ can capture the uncertainty in the areas that we have the support of offline data. However, no devoted argument can be made about $\mu_\theta$ and $\Sigma_\phi$ over out-of-distribution (OOD) data points, and thus, they might predict arbitrary values in those regions.

### 3.2.2 REGULARIZATION PHASE

If we directly employ the trained model from the warm-up phase as a surrogate for the actual environment and optimize a policy, the learned policy will most likely perform poorly in the real MDP (Levine et al., 2020). This happens because while training the policy, the RL algorithm will query the model in OOD data points as well, meaning that it will rely on potentially inaccurate predictions of the model, which might lead to inferior policies due to over- or under-estimation of the values. Nevertheless, this is not an inherent issue in standard model-based RL as the model can be improved over time by collecting more transitions, whereas in offline settings, the problem remains due to lack of additional interactions with the real environment. Therefore, we aim to find an efficient way to either prevent the policies from exploiting unreliable predictions, or prevent the unreliable predictions from affecting the values of state-action pairs.

To address this problem, we propose using a regularized objective function and a second phase for training the model. Once we have a preliminary empirical model, we expand the loss function to include a second objective to maximize the entropy of model predictions over the OOD domains. Since the model is characterized by a Gaussian distribution, entropy is a monotonically increasing function of $\det\left(\Sigma_\phi(s, a)\right) = \Sigma_i\Sigma_{\phi,i}(s, a)$ (Equation 1). Hence, higher entropy, which translates to higher uncertainty, results in a higher value for $\det\left(\Sigma_\phi(s, a)\right)$, and vice versa. As a result, the maximized entropy, which can be attained by maximizing $\det\left(\Sigma_\phi(s, a)\right)$, provides an upper-bound estimation on the uncertainty of model predictions.

Let $\mu_\theta(s, a)$ and $\Sigma_\phi(s, a)$ be the mean and covariance of the Gaussian distribution over the next state and reward, i.e., $P(s', r|s, a) = \mathcal{N}\left(\mu_\theta(s, a), \Sigma_\phi(s, a)\right)$, and $\bar{\mathbb{B}}_{\pi_e}$ be a sample batch of rollouts generated from the trained dynamics $\mu_\theta(s, a)$ using an exploration policy $\pi_e(a|s)$, i.e., $(s', r) = \mu_\theta\left(s, a \sim \pi_e(a|s)\right)$, the second loss $\mathcal{L}_2$ is thus defined as

$$\mathcal{L}_2(\phi; \bar{\mathbb{B}}_{\pi_e}) = \frac{1}{|\bar{\mathbb{B}}_{\pi_e}|} \sum_{(s,a,r,s') \in \bar{\mathbb{B}}_{\pi_e}} \sum_{i=1}^{d+1} -\frac{1}{2}\log\left(\Sigma_{\phi,i}(s, a)\right), \quad (3)$$

which together with $\mathcal{L}_1$ introduced in Equation 2 yield the hybrid loss $\mathcal{L}$

$$\mathcal{L}(\theta, \phi; \mathbb{B}_\mathcal{D}, \bar{\mathbb{B}}_{\pi_e}) = \mathcal{L}_1(\theta, \phi; \mathbb{B}_\mathcal{D}) + \alpha\mathcal{L}_2(\phi; \bar{\mathbb{B}}_{\pi_e}), \quad (4)$$

and $\alpha$ is a fixed constant to control the effect of regularization on the NLL loss.

Algorithm 1 summarizes the procedure to generate a sample batch of $b$ rollouts of length $h$ given an offline dataset $\mathcal{D}$, a trained model in the form of $\mathcal{N}\left(\mu_\theta(s, a), \Sigma_\phi(s, a)\right)$, and an exploration policy

---

**Algorithm 2** EMO

---

**Require:** Offline data $\mathcal{D}$, exploration policy $\pi_e$, batch size $b$, rollout horizon $h$, penalty coefficient $\lambda$, regularization coefficient $\alpha$
1: Initialize $\theta$ and $\phi$
2: **for** $K_1$ iterations **do** ▷ Warm-up phase
3:     Sample a batch of transitions $\mathbb{B}_{\mathcal{D}}$ from the offline dataset $\mathcal{D}$.
4:     Compute $\mathcal{L}_1(\theta, \phi; \mathbb{B}_{\mathcal{D}})$ (Equation 2.)
5:     Compute gradients and update $\theta$ and $\phi$.
6: **end for**
7: **for** $K_2$ iterations **do** ▷ Regularization phase
8:     Sample a batch of transitions $\mathbb{B}_{\mathcal{D}}$ from the offline dataset $\mathcal{D}$.
9:     Compute $\mathcal{L}_1(\theta, \phi; \mathbb{B}_{\mathcal{D}})$ (Equation 2.)
10:    Generate a batch of transitions $\bar{\mathbb{B}}_{\pi_e}$ using model rollouts (Algorithm 1.)
11:    Compute $\mathcal{L}_2(\phi; \bar{\mathbb{B}}_{\pi_e})$ (Equation 3.)
12:    Compute $\mathcal{L}(\theta, \phi; \mathbb{B}_{\mathcal{D}}, \bar{\mathbb{B}}_{\pi_e}) = \mathcal{L}_1(\theta, \phi; \mathbb{B}_{\mathcal{D}}) + \alpha \mathcal{L}_2(\phi; \bar{\mathbb{B}}_{\pi_e})$.
13:    Compute gradients and update $\theta$ and $\phi$.
14: **end for**

---

$\pi_e(a|s)$. In this algorithm, $\mu_\theta(s, a)$ models the transition dynamics and $\Sigma_\phi(s, a)$ is used as the uncertainty estimator. We leverage an exploratory policy as well as the dynamics model, which is initially trained in the warm-up phase, to generate rollouts additional to samples from offline data to be employed in the regularized optimization problem. Note that the dynamics model will be gradually updated as the training progresses. Accordingly, Algorithm 2 plots the overall learning procedure of EMO, in which the batches of data from both the offline dataset as well as the generated rollouts from the model are utilized to minimize the hybrid loss in Equation 4.

In the regularization phase, the model is trained to maximize the likelihood of the offline data, collected by a (possibly unknown) behavior policy, while maximizing the entropy of predictions over the distribution induced by an exploratory policy. Hence, $\mathcal{L}_1$ ensures that $\mu_\theta$ and $\Sigma_\phi$ maintain their accuracy in the support of offline data, while $\mathcal{L}_2$ aims to increase $\Sigma_\phi$ as we leave the support of offline dataset (over the distribution induced by the exploration policy $\pi_e$.) As a result, $\Sigma_\phi(s, a)$ can be used as an upper bound indicating how reliable/accurate the trained model is for a certain pair of $s$ and $a$, as long as the distribution of rollouts generated by the exploration policy covers this particular state-action pair. In other words, as long as the distribution of rollouts generated from $\pi_e$ covers the potential distributions of other exploratory policies, (which will be used later during policy optimization in Section 3.3,) $\Sigma_\phi(s, a)$ can be leveraged as the error estimator to relatively penalize unreliable predictions.

Furthermore, by considering a small value for $\alpha$, we ensure that the effect of $\mathcal{L}_2$ will be negligible compared to $\mathcal{L}_1$ where the distribution of offline data $\mathcal{D}$ overlaps with the distribution of rollouts under the exploratory policy $\pi_e$. In this way, the performance of the model is practically unaffected in the support of offline data and possibly other generalizable neighborhoods. Note that $\mathcal{L}_2$ will still be effective in the OOD areas, since $\mathcal{L}_1$ is non-existent in those regions.

---

**Algorithm 3** General Framework for Model-based Offline RL

---

**Require:** Offline dataset $\mathcal{D} = \{(s_i, a_i, r_i, s_i')\}_{i=1}^n$; penalty coefficient $\lambda$.
1: Train the dynamics model $\mu_\theta$ and admissible uncertainty estimator $\Sigma_\phi$ using $\mathcal{D}$. (Algorithm 2)
2: Define $u(s, a) = \sqrt{\det\left(\Sigma_\phi(s, a)\right)}$
3: Define empirical MDP $\hat{M}$ with dynamics $\mu_\theta$ and reward $\tilde{r}(s, a) = r(s, a) - \lambda u(s, a)$.
4: Run any RL algorithm on $\hat{M}$ until convergence to obtain $\hat{\pi}^* = \arg\max_\pi \eta_{\hat{M}}(\pi)$.

---

### 3.3 POLICY LEARNING

Once the model is trained using EMO, we will utilize its components to define a pessimistic MDP, which can be coupled with any policy optimization technique to obtain the output policy $\hat{\pi}^*$. Sub-

sequently, the overall learning framework, which also adheres to prior work (Yu et al., 2020; Kidambi et al., 2020), is summarized in Algorithm 3. In this algorithm, the offline dataset $\mathcal{D}$ is first used to train a dynamics model $\mu_\theta$ along with an admissible error/uncertainty estimator $\Sigma_\phi$. Next, a new pessimistic MDP is defined as $\hat{M} = (\mathcal{S}, \mathcal{A}, T_\mu, \tilde{r}, \gamma, \rho_0)$, with $T_\mu = \{\mu_{\theta,i}(s,a)\}_{i=1}^d$ and $\tilde{r}(s,a) = r(s,a) - \lambda u(s,a)$, where $u(s,a) = \sqrt{\mathrm{tr}(\Sigma_\phi(s,a))}$ (see section 3.2). Lastly, the pessimistic MDP $\hat{M}$ is leveraged as a surrogate model to train an RL algorithm and obtain $\hat{\pi}^*$.

## 3.4 THEORETICAL GROUNDS OF EMO

We argue that EMO is an extension to MOPO (Yu et al., 2020), while addresses the limitations of MOPO regarding learning an ensemble of models for uncertainty estimation. Accordingly, we expand the theoretical grounds of MOPO to EMO, where we guarantee conservative policy evaluation and safe policy improvement of EMO, regardless of the stochasticity of the environment. For a detailed discussion on the theoretical analysis, refer to Appendix A.1.

## 4 EMPIRICAL STUDY

In this section, we evaluate the performance of our approach on D4RL benchmark datasets for MuJoCo environments (Fu et al., 2020). We include data from three different environments: halfcheetah, hopper, and walker2d, and four different types from each, i.e., random, medium, medium-replay, and medium-expert, which results in twelve different datasets. Throughout the experiments, we aim to investigate the following questions: (1) How does EMO perform compared to SOTA methods on standard offline RL benchmark? (2) What impact do entropy regularization and reward penalty have on the performance of the trained policies? (3) What is the effect of exploration policy $\pi_e$ on the generalization ability of the trained models and the performance of the resulting policies? (4) What is the effect of regularization coefficient $\alpha$ on the performance of EMO?

## 4.1 EXPERIMENTAL SETUPS

Following the setup in Yu et al. (2020), we characterize the model as a 4-layer feed-forward neural network across all domains, with 200 hidden units in each layer. Subsequently, the output of the last hidden layer is fed into a two-head network architecture to generate $\mu_\theta(s,a)$ and $\Sigma_\phi(s,a)$, where $\mu_\theta$ and $\Sigma_\phi$ are two outputs of a single neural network, i.e., they share the same network, except for their output layers. Instead of directly estimating the reward function, the model predicts the center of mass velocity, and the reward is calculated afterwards based on its formulation in each domain.

For all the experiments, a soft actor-critic (SAC) agent[1] (Haarnoja et al., 2018) is used as the reinforcement learning agent for policy optimization on the trained pessimistic MDP $\hat{M}$. At each time step, a batch of $k$-step rollouts are generated and added to the replay memory of the SAC agent, where the actions for generating the rollouts are taken based on the current policy of the agent, while transitions and rewards are produced by $\hat{M}$. Next, the agent optimizes its policy based on samples from both its replay memory and the offline data $\mathcal{D}$. Note that in our implementation, the agent will only utilize its own replay memory for policy optimization, meaning that samples from offline data are not directly utilized for policy optimization (see Algorithm 5 in Appendix A.5). Consequently, the resulting policies from SAC are evaluated in the real MuJoCo environment for testing.

## 4.2 OVERALL PERFORMANCE

We compare the performance of EMO to SOTA model-based offline algorithms, i.e., MOPO (Yu et al., 2020), MOReL (Kidambi et al., 2020), COMBO (Yu et al., 2021), RAMBO-RL (Rigter et al., 2022), and GELATO (Tennenholtz et al., 2021), as well as 3 model-free counterparts, i.e., UWAC (Wu et al., 2021), CQL (Kumar et al., 2020), and ATAC (Cheng et al., 2022). The performance results of both model-based and model-free techniques are summarized in Table 1. All the presented scores are normalized according to the procedure proposed in Fu et al. (2020). For EMO, the results are the performance of policy at the last iteration of training, averaged over 3 random seeds $\pm$

---

[1]https://github.com/pranz24/pytorch-soft-actor-critic

Table 1: Performance results of offline RL algorithms on D4RL benchmark datasets.

| Dataset | Environment | EMO (Ours) | RAMBO-RL | COMBO | MOPO | MOReL | UWAC | CQL | ATAC | GELATO |
|---------|-------------|------------|----------|-------|------|-------|------|-----|------|--------|
| random | halfcheetah | $36.3 \pm 1.7$ | $\mathbf{39.5} \pm 1.1$ | $38.8 \pm 1.5$ | 35.4 | 25.6 | 14.5 | 35.4 | 3.9 | 21.1 |
| random | hopper | $31.6 \pm 0.1$ | $25.4 \pm 2.4$ | $17.9 \pm 0.6$ | 11.7 | $\mathbf{53.6}$ | 22.4 | 10.8 | 17.5 | 21.2 |
| random | walker2d | $5.3 \pm 2.9$ | $0.0 \pm 0.1$ | $7.0 \pm 1.5$ | 13.6 | $\mathbf{37.3}$ | 15.5 | 7.0 | 6.8 | 9.0 |
| medium | halfcheetah | $68.5 \pm 0.9$ | $\mathbf{77.9} \pm 1.1$ | $54.2 \pm 0.6$ | 42.3 | 42.1 | 46.5 | 44.4 | 53.3 | 42.6 |
| medium | hopper | $36.5 \pm 25.1$ | $87.0 \pm 4.9$ | $\mathbf{97.2} \pm 0.9$ | 28.0 | 95.4 | 88.9 | 86.6 | 85.6 | 51.8 |
| medium | walker2d | $84.1 \pm 2.5$ | $84.9 \pm 0.8$ | $81.9 \pm 1.1$ | 17.8 | 77.8 | 57.5 | 79.2 | $\mathbf{89.6}$ | 27.6 |
| medium-replay | halfcheetah | $56.7 \pm 5.9$ | $\mathbf{68.7} \pm 1.7$ | $55.1 \pm 0.4$ | 53.1 | 40.2 | 46.8 | 46.2 | 48.0 | - |
| medium-replay | hopper | $77.4 \pm 16.9$ | $99.5 \pm 1.5$ | $89.5 \pm 0.7$ | 67.5 | 93.6 | 39.4 | 48.6 | $\mathbf{102.5}$ | - |
| medium-replay | walker2d | $83.6 \pm 1.3$ | $89.2 \pm 2.1$ | $56.0 \pm 3.5$ | 39.0 | 49.8 | 27.0 | 26.7 | $\mathbf{92.5}$ | - |
| medium-expert | halfcheetah | $80.2 \pm 3.9$ | $95.4 \pm 1.7$ | $90.0 \pm 2.3$ | 63.3 | 53.3 | $\mathbf{127.4}$ | 62.4 | 94.8 | 65.8 |
| medium-expert | hopper | $92.1 \pm 2.8$ | $88.2 \pm 6.5$ | $111.1 \pm 1.2$ | 23.7 | 108.7 | $\mathbf{134.7}$ | 111 | 111.9 | 17.7 |
| medium-expert | walker2d | $110.8 \pm 3.0$ | $56.7 \pm 12.3$ | $103.3 \pm 2.3$ | 44.6 | 95.6 | 99.7 | 98.7 | $\mathbf{114.2}$ | 33.0 |

standard error. The results for CQL are taken from D4RL benchmark white-paper (Fu et al., 2020). As for the values of other methods, the result are taken from their respective papers.

The outlined results in Table 1 demonstrate that EMO outperforms MOPO and GELATO in almost all cases, which shows that EMO, as an extension to MOPO, clearly improves upon its predecessor by replacing ensemble uncertainty quantification with entropy regularization. Furthermore, EMO achieves competitive results compared to COMBO and MOReL, outperforming both on 5 out of 12 datasets, which places EMO among the highest-performing model-based algorithms. Although EMO can achieve comparable results to RAMBO-RL in certain scenarios, its performance falls short in some cases, and only outperforms RAMBO-RL on one dataset. This can be attributed to the fact that RAMBO-RL is a task-specific method, which tunes the model to be pessimistic with respect to the current policy of the agent, while EMO utilizes a general purpose, task-agnostic model of the environment for policy optimization. In addition, bear in mind that EMO achieves this level of performance using only a single model of the environment, while an ensemble of models is utilized in other model-based methods. Moreover, Table 1 illustrates that EMO outperforms CQL and UWAC techniques in 8 out of 12 datasets, and performs competitively against ATAC, while outperforming ATAC on 4 out of 12 datasets, which highlights the effectiveness of our simple, efficient method in keeping up with the SOTA baselines.

Table 2: Performance results of policy optimization on different model configurations

| Dataset | Environment | EMO | NLL+SAC ($\lambda \neq 0$) | NLL+SAC ($\lambda = 0$) |
|---------|-------------|-----|-----------------------------|--------------------------|
| medium | halfcheetah | $\mathbf{68.5} \pm 0.9$ | $35.1 \pm 8.0$ | $0.9 \pm 0.9$ |
| medium | walker2d | $\mathbf{84.1} \pm 2.5$ | $13.0 \pm 13.0$ | $5.6 \pm 5.1$ |
| medium-replay | halfcheetah | $\mathbf{56.7} \pm 5.9$ | $4.0 \pm 3.8$ | $10.8 \pm 1.6$ |
| medium-replay | walker2d | $\mathbf{83.6} \pm 1.3$ | $18.7 \pm 15.8$ | $6.4 \pm 1.2$ |

## 4.3 ABLATION STUDIES

In order to address question (2), we conduct an experiment to compare the performance of EMO with two simplified versions of the algorithm: (i) training the policy on a model that is only trained via the NLL loss (i.e., the resulting model from the warm-up phase) *without* utilizing the reward penalty ($\lambda = 0$), denoted by **NLL+SAC** ($\lambda = 0$), and (ii) the same model *with* the reward penalty ($\lambda \neq 0$) leveraging the covariance matrix from the warm-start phase, indicated by **NLL+SAC** ($\lambda \neq 0$). The outcomes of the experiment are outlined in Table 2 on 4 different datasets. The results confirm that although penalizing the reward values in OOD areas improves the performance of the model, it is still insufficient to achieve comparable results to state-of-the-arts. The reason lies in the fact that the penalties depend on the arbitrary predictions of the covariance matrix in the OOD domains. However, by incorporating the entropy regularization step, we ensure that reward penalties will be proportional to the associated uncertainty of the predictions, which leads to a considerable gap between the performance of the resulting policies. This finding empirically validates the impact of the entropy regularization coupled with penalizing the reward values in uncertain regions.

In the second ablation study, we investigate the impact of using an informed exploration policy vs. a random policy to address question (3). One could argue that for a specific task, utilizing an informed policy (i.e., the current policy of the agent) instead of a random policy can be beneficial. In this case,

Table 3: Performance results of policy optimization for random vs. informed exploration policy

| Dataset | Environment | EMO | Modified EMO | RAMBO-RL |
|---------|-------------|-----|--------------|----------|
| medium-replay | halfcheetah | $56.7 \pm 5.9$ | $\mathbf{61.1} \pm 0.4$ | $\mathbf{68.7} \pm 1.7$ |
| medium-replay | walker2d | $83.6 \pm 1.3$ | $\mathbf{85.5} \pm 3.5$ | $\mathbf{89.2} \pm 2.1$ |

the entropy regularization phase can be guided more efficiently toward the areas that are more likely to be explored by the learning agent. Subsequently, the coverage of the rollouts generated under an informed policy is expected to be a small subset of the rollouts generated from a random policy.

Consequently, we propose a modified version of EMO, summarized in Algorithm 4 in Appendix A.5, in which the (SAC) agent and the model (only the regularization phase) are trained simultaneously, rather than modular. In this variant, the model utilizes the current policy of the agent as the exploration policy for entropy regularization in order to benefit from an informed exploration scheme. We further examine this modification on two offline datasets, namely halfcheetah-medium-replay and walker2d-medium-replay, compared to RAMBO-RL. The summarized results in Table 3 indicate that informed exploration can indeed improve the performance of trained policies. Such improvements in the performance are expected since the modified version of EMO has a more specialized, task-specific approach for model optimization compared to the original version. Furthermore, although EMO can achieve comparable results compared to RAMBO-RL and even outperform it on one set of data (see Table 1), the modified version can tighten the gap and make the results more competitive to RAMBO-RL. However, we still prioritize the original EMO, as it has general purpose, task-agnostic pessimistic model of the environment, which can be employed in any modular framework with any RL algorithm for training policies. Whereas this cannot be achieved using the modified version of EMO or methods such as RAMBO-RL.

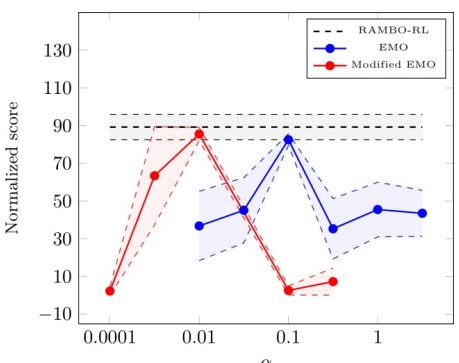

Figure 2: Performance of EMO variants in terms of $\alpha$ on walker2d-medium-replay.

To answer question (4), we demonstrate the performance of EMO and modified EMO for different configurations of the regularization coefficient $\alpha$ on walker2d-medium-replay dataset in figure 2. The figure illustrates how $\alpha$ affects the performance of both methods, also compared to RAMBO-RL. As we increase $\alpha$, the regularization term in Equation 4 becomes more dominant, leading to more conservative algorithms, and vice versa. As a result, smaller values of $\alpha$ are preferred to achieve best performance when offline data has limited coverage of the state-action space or contains less informative data. Conversely, when the offline data provides a better coverage, higher values for $\alpha$ are preferred. In addition, since the datastes from the medium-replay category are considerably smaller than other D4RL benchmark data (aka limited coverage), both EMO and modified EMO are expected to perform better when $\alpha$ is set to smaller numbers, as shown in figure 2. However, if $\alpha$ is too small, then the algorithms will not regularize very well, and that can lead a decrease in performance as well. The same can be said when the algorithm becomes too conservative, which is the case when $\alpha$ is set too large.

## 5 CONCLUSIONS

In this paper, we presented EMO, an entropy-regularized optimization algorithm to learn a pessimistic MDP for model-based offline RL problems. In this framework, we devised a hybrid loss function to minimize the NLL of the model on the distribution of offline data while maximizing the entropy over OOD domains. We thus optimized both objectives in a single model rather than an ensemble of models as in SOTA model-based approaches. Moreover, our empirical study on D4RL benchmark data showed that our approach competes with SOTA offline RL techniques.

## 6 REPRODUCIBILITY STATEMENT

To ensure the reproducibility of the results, the codes are provided in the supplementary materials. Configurations and the choice of hyperparameters are also included in the supplementary materials in `readme.txt` file. To facilitate the understanding of theoretical formulation and practical implementation, additional algorithms and theories are also included in the appendix.

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

# A APPENDIX

## A.1 THEORETICAL GROUNDS OF EMO

As discussed in Section 3.4, EMO can be considered as an extension to MOPO (Yu et al., 2020), with the aim to address the limitations of ensemble uncertainty estimation. Accordingly, in this section, we expand upon Lemma 4.1 from MOPO in order to establish the theoretical grounds for EMO.

### A.1.1 PRELIMINARIES

We specify a Markov decision process (MDP) by the tuple $M = (\mathcal{S}, \mathcal{A}, T, r, \mu_0, \gamma)$, where $\mathcal{S}$ and $\mathcal{A}$ denote the state space and action space, respectively, $T(s'|s, a)$ is the transition dynamics, $r(s, a)$ is the reward function, $\gamma \in (0, 1)$ is the discount factor, and $\mu_0$ is the distribution of the initial state. A policy $\pi(a|s)$ is defined as a mapping from states to a distribution over actions $\pi : \mathcal{S} \times \mathcal{A} \to [0, 1]$. If we define $\mathbb{P}(s_t = s|\mu_0, \pi, T)$ as the probability of being in state $s$ at time step $t$ when following policy $\pi$ from an initial state sampled from $\mu_0$, in an environment with transition dynamics $T$, then the discounted state-action visitation distribution of policy $\pi$ under dynamics $T$ can be defined as $\rho_T^\pi(s, a) = (1 - \gamma)\pi(a|s)\sum_{t=0}^{\infty} \gamma^t \mathbb{P}(s_t = s|\mu_0, \pi, T)$. The goal is to learn a policy that maximizes the expected discounted return $\eta_M(\pi)$ when followed: $\max_\pi \eta_M(\pi) := \frac{1}{1-\gamma}\mathbb{E}_{(s,a)\sim\rho_T^\pi(s,a)}[r(s, a)]$. The value function, $V_M^\pi(s) = \mathbb{E}_{\pi,T}[\Sigma_{t=0}^{\infty}\gamma^t r(s_t, a_t)|s_0 = s]$, is regarded as the value of a particular state $s$ under the policy $\pi$, which is defined as the expected discounted return under $\pi$ when starting from $s$.

In offline RL framework, we only have access to a static dataset of transition tuples $\mathcal{D} = \{(s, a, r, s')\}$, which is collected by running a behavior policy $\pi^B$ in the real environment. In offline RL setting, the goal is to find the best possible policy using the static offline dataset $\mathcal{D}$.

### A.1.2 THEORETICAL FORMULATION

We start by expanding the theoretical formulation of EMO and proving that EMO can guarantee conservative policy evaluation and safe policy improvement. First, we quantify the relationship between the performance of a policy $\pi$ under two arbitrary MDPs. From the theoretical formulation of MOPO (Yu et al., 2020), we have:

**Lemma A.1.1** (Telescoping lemma). *Let $M$ and $\hat{M}$ be two MDPs with the same reward function $r$, but different dynamics $T$ and $\hat{T}$ respectively. For any arbitrary policy $\pi$, let $G_{\hat{M}}^\pi(s, a) := \mathbb{E}_{s'\sim\hat{T}(s,a)}[V_M^\pi(s')] - \mathbb{E}_{s'\sim T(s,a)}[V_M^\pi(s')]$. Then,*

$$\eta_{\hat{M}}(\pi) - \eta_M(\pi) = \frac{\gamma}{1-\gamma}\mathbb{E}_{(s,a)\sim\rho_{\hat{T}}^\pi(s,a)}[G_{\hat{M}}^\pi(s, a)] \tag{5}$$

Next, we expand on Lemma A.1.1 to include MDPs with different rewards as well.

**Lemma A.1.2.** *Let $M$ and $\hat{M}$ be two MDPs with reward functions $r$ and $\hat{r}$, and transition dynamics $T$ and $\hat{T}$ respectively. For any arbitrary policy $\pi$, let $G_{\hat{M}}^\pi(s, a) := \mathbb{E}_{s'\sim\hat{T}(s,a)}[V_M^\pi(s')] - \mathbb{E}_{s'\sim T(s,a)}[V_M^\pi(s')]$, and $\hat{r}(s, a) - r(s, a) = \delta r(s, a)$. Then,*

$$\eta_{\hat{M}}(\pi) - \eta_M(\pi) = \frac{1}{1-\gamma}\mathbb{E}_{(s,a)\sim\rho_{\hat{T}}^\pi(s,a)}[\delta r(s, a) + \gamma G_{\hat{M}}^\pi(s, a)] \tag{6}$$

*Proof.* Considering $\hat{r}(s, a) = r(s, a) + \delta r(s, a)$, we have

$$\eta_{\hat{M}}(\pi) = \frac{1}{1-\gamma}\mathbb{E}_{(s,a)\sim\rho_{\hat{T}}^\pi(s,a)}[r(s, a)] + \frac{1}{1-\gamma}\mathbb{E}_{(s,a)\sim\rho_{\hat{T}}^\pi(s,a)}[\delta r(s, a)] \tag{7}$$

The first term in the RHS of Equation 7 corresponds to the return of the policy under dynamics $\hat{T}$ and reward $r$. According to Lemma A.1.1, we have

$$\frac{1}{1-\gamma}\mathbb{E}_{(s,a)\sim\rho_{\hat{T}}^\pi(s,a)}[r(s,a)] = \eta_M(\pi) + \frac{\gamma}{1-\gamma}\mathbb{E}_{(s,a)\sim\rho_{\hat{T}}^\pi(s,a)}[G_{\hat{M}}^\pi(s,a)] \tag{8}$$

Now, by substituting Equation 8 into Equation 7, we get

$$\begin{aligned}
\eta_{\hat{M}}(\pi) - \eta_M(\pi) &= \frac{\gamma}{1-\gamma}\mathbb{E}_{(s,a)\sim\rho_{\hat{T}}^\pi(s,a)}[G_{\hat{M}}^\pi(s,a)] + \frac{1}{1-\gamma}\mathbb{E}_{(s,a)\sim\rho_{\hat{T}}^\pi(s,a)}[\delta r(s,a)] \\
&= \frac{1}{1-\gamma}\mathbb{E}_{(s,a)\sim\rho_{\hat{T}}^\pi(s,a)}[\delta r(s,a) + \gamma G_{\hat{M}}^\pi(s,a)]
\end{aligned} \tag{9}$$

$\square$

Lemma A.1.2 quantifies the difference between the return of a policy under two arbitrary MDPs, as long as they share the same $(\mathcal{S}, \mathcal{A}, \mu_0, \gamma)$. For the specific case of EMO, we can consider $M$ as the actual MDP, and $\hat{M}$ as the learned MDP on the offline data, based on EMO algorithm, *before* applying the reward penalty. For now, we model the penalized reward function by $\tilde{r}(s,a) = \hat{r}(s,a) - \lambda\Delta r(s,a)$, where $\lambda\Delta r(s,a)$ is the penalty we apply to the learned reward, such that $\lambda \geq 0$. Let $\tilde{M} = (\mathcal{S}, \mathcal{A}, \hat{T}, \tilde{r}, \mu_0, \gamma)$ denote the MDP with transition dynamics $\hat{T}$ and reward function $\tilde{r}(s,a)$. We have

$$\begin{aligned}
\eta_{\tilde{M}}(\pi) &= \frac{1}{1-\gamma}\mathbb{E}_{(s,a)\sim\rho_{\hat{T}}^\pi(s,a)}[\tilde{r}(s,a)] \\
&= \frac{1}{1-\gamma}\mathbb{E}_{(s,a)\sim\rho_{\hat{T}}^\pi(s,a)}[\hat{r}(s,a)] - \frac{\lambda}{1-\gamma}\mathbb{E}_{(s,a)\sim\rho_{\hat{T}}^\pi(s,a)}[\Delta r(s,a)]
\end{aligned} \tag{10}$$

The first term in the RHS of Equation 10 corresponds to the return of the policy under dynamics $\hat{T}$ and reward $\hat{r}$, namely $\eta_{\hat{M}}(\pi)$. Using Lemma A.1.2, we can rewrite this term as

$$\eta_{\hat{M}}(\pi) = \frac{1}{1-\gamma}\mathbb{E}_{(s,a)\sim\rho_{\hat{T}}^\pi(s,a)}[\hat{r}(s,a)] = \eta_M(\pi) + \frac{1}{1-\gamma}\mathbb{E}_{(s,a)\sim\rho_{\hat{T}}^\pi(s,a)}[\delta r(s,a) + \gamma G_{\hat{M}}^\pi(s,a)] \tag{11}$$

By substituting 11 in 10 we have

$$\eta_{\tilde{M}}(\pi) - \eta_M(\pi) = \frac{1}{1-\gamma}\mathbb{E}_{(s,a)\sim\rho_{\hat{T}}^\pi(s,a)}[\delta r(s,a) + \gamma G_{\hat{M}}^\pi(s,a)] - \frac{\lambda}{1-\gamma}\mathbb{E}_{(s,a)\sim\rho_{\hat{T}}^\pi(s,a)}[\Delta r(s,a)] \tag{12}$$

In order to achieve conservative policy evaluation, we need to ensure that the performance of any policy is not overestimated under the penalized MDP $\tilde{M}$. Thus, we need to ensure that $\eta_{\tilde{M}}(\pi) - \eta_M(\pi) \leq 0$ for all $\pi$.

**Proposition A.1.** *The penalized MDP $\tilde{M}$ preserves conservative policy evaluation if,*

$$\lambda\mathbb{E}_{(s,a)\sim\rho_{\hat{T}}^\pi(s,a)}[\Delta r(s,a)] \geq \mathbb{E}_{(s,a)\sim\rho_{\hat{T}}^\pi(s,a)}[\delta r(s,a) + \gamma G_{\hat{M}}^\pi(s,a)], \forall\pi\in\Pi \tag{13}$$

Proposition A.1 establishes the condition on the adjustment $\lambda\Delta r(s,a)$ to guarantee conservative policy evaluation. Once again, note that in order to preserve conservative policy evaluation in the penalized MDP $\tilde{M}$, the condition stated in 13 should hold for any arbitrary policy $\pi$. A direct implication of Proposition A.1 is that training any policy in $\tilde{M}$ is equal to optimizing a lower bound on the return under the real MDP $M$.

**Practical Implication.** If we can ensure $\mathbb{E}_{(s,a)\sim\rho_{\hat{T}}^\pi(s,a)}[\Delta r(s,a)] \geq 0, \forall\pi\in\Pi$, we can satisfy the condition on conservative policy evaluation by choosing $\lambda$ large enough, assuming that $\delta r(s,a)$ and $G_{\hat{M}}^\pi(s,a)$ are bounded.

In addition to conservative policy evaluation, we want to guarantee safe policy improvement over the behavior policy $\pi^B$ as well. Let $\tilde{\pi}$ be the optimal policy trained on the penalized MDP $\tilde{M}$. We first quantify the performance difference between $\pi^B$ and $\tilde{\pi}$ under the actual MDP $M$:

$$
\begin{aligned}
&\eta_M(\tilde{\pi}) - \eta_M(\pi^B) \\
&\overset{(12)}{=} \eta_{\tilde{M}}(\tilde{\pi}) - \frac{1}{1-\gamma}\mathbb{E}_{(s,a)\sim\rho_{\hat{T}}^{\tilde{\pi}}(s,a)}[\delta r(s,a) + \gamma G_{\hat{M}}^{\tilde{\pi}}(s,a)] + \frac{\lambda}{1-\gamma}\mathbb{E}_{(s,a)\sim\rho_{\hat{T}}^{\tilde{\pi}}(s,a)}[\Delta r(s,a)] \\
&\quad -\eta_{\tilde{M}}(\pi^B) + \frac{1}{1-\gamma}\mathbb{E}_{(s,a)\sim\rho_{\hat{T}}^{\pi^B}(s,a)}[\delta r(s,a) + \gamma G_{\hat{M}}^{\pi^B}(s,a)] - \frac{\lambda}{1-\gamma}\mathbb{E}_{(s,a)\sim\rho_{\hat{T}}^{\pi^B}(s,a)}[\Delta r(s,a)] \\
&= \eta_{\tilde{M}}(\tilde{\pi}) - \eta_{\tilde{M}}(\pi^B) - \frac{1}{1-\gamma}\mathbb{E}_{(s,a)\sim\rho_{\hat{T}}^{\tilde{\pi}}(s,a)}[\delta r(s,a) + \gamma G_{\hat{M}}^{\tilde{\pi}}(s,a)] \\
&\qquad\qquad + \frac{1}{1-\gamma}\mathbb{E}_{(s,a)\sim\rho_{\hat{T}}^{\pi^B}(s,a)}[\delta r(s,a) + \gamma G_{\hat{M}}^{\pi^B}(s,a)] \\
&\qquad\qquad + \frac{\lambda}{1-\gamma}(\mathbb{E}_{(s,a)\sim\rho_{\hat{T}}^{\tilde{\pi}}(s,a)}[\Delta r(s,a)] - \mathbb{E}_{(s,a)\sim\rho_{\hat{T}}^{\pi^B}(s,a)}[\Delta r(s,a)])
\end{aligned}
$$

Note that since $\tilde{\pi}$ is the optimal policy under the penalized MDP $\tilde{M}$, then we have $\eta_{\tilde{M}}(\tilde{\pi}) - \eta_{\tilde{M}}(\pi^B) = C_{\tilde{M}}(\pi^B) \geq 0$.

**Proposition A.2.** *Let $\tilde{\pi}(a|s)$ be the optimal policy trained on the penalized MDP $\tilde{M}$. Then, $\tilde{\pi}$ is a safe policy improvement over $\pi^B$, i.e. $\eta_M(\tilde{\pi}) \geq \eta_M(\pi^B)$, if*

$$
\begin{aligned}
\lambda(\mathbb{E}_{(s,a)\sim\rho_{\hat{T}}^{\tilde{\pi}}(s,a)}[\Delta r(s,a)] - &\mathbb{E}_{(s,a)\sim\rho_{\hat{T}}^{\pi^B}(s,a)}[\Delta r(s,a)]) \\
&\geq \mathbb{E}_{(s,a)\sim\rho_{\hat{T}}^{\tilde{\pi}}(s,a)}[\delta r(s,a) + \gamma G_{\hat{M}}^{\tilde{\pi}}(s,a)] \\
&\quad -\mathbb{E}_{(s,a)\sim\rho_{\hat{T}}^{\pi^B}(s,a)}[\delta r(s,a) + \gamma G_{\hat{M}}^{\pi^B}(s,a)] - (1-\gamma)C_{\tilde{M}}(\pi^B)
\end{aligned} \tag{14}
$$

*which satisfies*

$$
\begin{aligned}
&\eta_M(\tilde{\pi}) - \eta_M(\pi^B) \\
&= C_{\tilde{M}}(\pi^B) - \frac{1}{1-\gamma}\mathbb{E}_{(s,a)\sim\rho_{\hat{T}}^{\tilde{\pi}}(s,a)}[\delta r(s,a) + \gamma G_{\hat{M}}^{\tilde{\pi}}(s,a)] \\
&\qquad + \frac{1}{1-\gamma}\mathbb{E}_{(s,a)\sim\rho_{\hat{T}}^{\pi^B}(s,a)}[\delta r(s,a) + \gamma G_{\hat{M}}^{\pi^B}(s,a)] \\
&\qquad + \frac{\lambda}{1-\gamma}(\mathbb{E}_{(s,a)\sim\rho_{\hat{T}}^{\tilde{\pi}}(s,a)}[\Delta r(s,a)] - \mathbb{E}_{(s,a)\sim\rho_{\hat{T}}^{\pi^B}(s,a)}[\Delta r(s,a)])
\end{aligned} \tag{15}
$$

Proposition A.2 establishes the condition on the penalty $\lambda \Delta r(s,a)$ to guarantee safe policy improvement over $\pi^B$ in the form of Inequality 14, and quantifies the improvement over $\pi^B$ in the form of Equation 15.

**Practical Implication.** If we can ensure $\mathbb{E}_{(s,a)\sim\rho_{\hat{T}}^{\tilde{\pi}}(s,a)}[\Delta r(s,a)] \geq \mathbb{E}_{(s,a)\sim\rho_{\hat{T}}^{\pi^B}(s,a)}[\Delta r(s,a)]$, we can satisfy the condition for safe policy improvement by choosing $\lambda$ large enough, assuming that $\delta r(s,a)$ and $G_{\hat{M}}^{\pi}(s,a)$ are bounded. Another similar yet more practical approach could be to ensure that $\mathbb{E}_{(s,a)\sim\rho_{\hat{T}}^{\pi}(s,a)}[\Delta r(s,a)] \geq \mathbb{E}_{(s,a)\sim\rho_{\hat{T}}^{\pi^B}(s,a)}[\Delta r(s,a)]$ for all $\pi \in \Pi$, which subsumes the original condition.

**Conservative policy evaluation of EMO.** In EMO, we define the penalty term as a positive, increasing function of the entropy $\Delta r(s,a) = u(s,a) = \sqrt{\det(\Sigma_\phi(s,a))}$. Thus, we have $\Delta r(s,a) \geq 0$ for all $(s,a) \in \mathcal{S} \times \mathcal{A}$, and it is obvious that $\mathbb{E}_{(s,a)\sim\rho_{\hat{T}}^{\pi}(s,a)}[\Delta r(s,a)] \geq 0$ for all $\pi \in \Pi$; as a result, according to the practical implications of Proposition A.1, conservative policy optimization can be achieved by choosing $\lambda$ large enough.

**Safe policy improvement of EMO.** Although we cannot theoretically guarantee safe policy improvement for $\Delta r(s,a) = \sqrt{\det(\Sigma_\phi(s,a))}$ (which is also the case for MOPO Yu et al. (2020), MOReL Kidambi et al. (2020), and COMBO Yu et al. (2021)), it is reasonable to assume that in practice, regardless of the stochasticity of the environment, $\Delta r(s,a)$ is smaller over $\rho_{\hat{T}}^{\pi^B}(s,a)$ than $\rho_{\hat{T}}^{\pi}(s,a)$ for any arbitrary $\pi$. Note that throughout the regularization phase of EMO, we specifically try to maximize the entropy over the OOD domains, and as a result, we expect lower entropy over the support of data, which we can assume has a distribution very close to $\rho_{\hat{T}}^{\pi^B}(s,a)$. By defining the penalty term as a positive, increasing function of the entropy, we can practically assume that $\mathbb{E}_{(s,a)\sim\rho_{\hat{T}}^{\pi}(s,a)}[\Delta r(s,a)] \geq \mathbb{E}_{(s,a)\sim\rho_{\hat{T}}^{\pi^B}(s,a)}[\Delta r(s,a)]$ for all $\pi \in \Pi$, and guarantee safe policy improvement according to practical implications of Proposition A.2, by choosing $\lambda$ large enough.

Thus, EMO in its original formulation, can be applied to any environment, whether deterministic or stochastic, and guarantee conservative policy evaluation and safe policy improvement.

### A.2 DIFFERENTIATION BETWEEN ALEATORIC AND EPISTEMIC UNCERTAINTY AND APPLICABILITY TO STOCHASTIC ENVIRONMENTS

Our model does not explicitly differentiate between aleatoric and epistemic uncertainties over OOD domains, and penalizes the rewards based on an upper bound over total uncertainty, instead of only epistemic uncertainty. However, we argue that this is a good approach in practice, even in the case of stochastic environments. In general, uncertainty, be it aleartoric or epistemic, is a source of error in RL algorithms. If we have a reliable and accurate estimation of either of these uncertainties, we can calculate their effect in our evaluations and algorithms. Otherwise, we should find ways to indirectly account for these sources of error, in order to prevent our methods from exploiting potential errors caused by these sources (e.g. by forming performance lower bounds with reward penalties etc.).

Same as any other measure, we argue that aleatoric uncertainty cannot be reliably quantified over OOD domains either, especially when we only have a single model; even if we have an ensemble of models, where we can estimate the aleatoric uncertainty by averaging over the variances of each model (Depeweg et al., 2018), the estimation would be inaccurate, as in practice, the sample size from the posterior distribution of parameters (the number of models in the ensemble) is small and samples are potentially correlated for the reasons discussed in the paper. Thus, aleatoric uncertainty over OOD domains becomes an unmeasurable source of error itself. As a result, it is reasonable to penalize a measure of total uncertainty over OOD domains, rather than only penalizing epistemic uncertainty, as we cannot confidently rely on the estimated aleatoric uncertainty over such domains. As for the in-distribution aleatoric uncertainty, we will show that our method preserves the reliable estimation of in-distribution aleatoric uncertainty. For this, we will discuss the characteristics of a model trained based on EMO over the support of data as well as OOD domains. We first restate the formulation of the proposed hybrid loss in Equation 4:

$$\mathcal{L}(\theta, \phi; \mathbb{B}_\mathcal{D}, \bar{\bar{\mathbb{B}}}_{\pi_e}) = \mathcal{L}_1(\theta, \phi; \mathbb{B}_\mathcal{D}) + \alpha \mathcal{L}_2(\phi; \bar{\bar{\mathbb{B}}}_{\pi_e})$$

where $\mathcal{L}_1$ corresponds to NLL loss, $\mathcal{L}_2$ corresponds to entropy regularization term, and $\alpha$ is the regularization coefficient. Please note that as discussed in Section 3.2.2 of the paper, we assume $\alpha$ has a small value.

(i) Since we assume that $\alpha$ is small, hybrid loss will be dominated by the NLL loss ($\mathcal{L}_1$) over the support of offline data. As a result, $\Sigma_\phi(s,a)$ will actually correspond to the aleatoric uncertainty over those regions (which, in the case of deterministic environments, is close to zero). Note that this estimation is reliable since it is learned over the support of offline data.

(ii) Over OOD domains, NLL loss ($\mathcal{L}_1$) does not exist (as we do not have any supervised data over OOD domains), and hybrid loss will be dominated by entropy regularization term. Thus, $\Sigma_\phi(s,a)$ will correspond to an upper bound of total uncertainty/error over OOD domains.

As a result, the estimated aleatoric uncertainty over the in-distribution data, is preserved by $\Sigma_\phi(s,a)$, as discussed in (i). We also present in Table 4 the average value of $\sqrt{\det(\Sigma_\phi(s,a))}$ over the samples drawn from the offline dataset for: (1) a model trained based on EMO; and (2) an NLL model, which goes to show that the uncertainty quantification is practically unaffected over in-distribution data,

meaning that the reliable aleatoric uncertainty is preserved in a model trained by EMO, which can be accounted for with minimal modifications to the original EMO algorithm, e.g. by excluding it from the penalty term.

| Environment | Dataset | EMO | NLL |
|---|---|---|---|
| walker2d | medium | 1.340 | 1.414 |
| walker2d | medium-replay | 2.694 | 2.792 |
| walker2d | medium-expert | 1.273 | 1.378 |

Table 4: the average value of $\sqrt{\det(\Sigma_\phi(s,a))}$ over the samples drawn from the offline dataset $\mathcal{D}$ for: (1) a model trained based on EMO; and (2) an NLL model.

Still, we would like to mention that EMO in its original form proposed in this paper, theoretically guarantees conservative policy evaluation and safe policy improvement, regardless of the stochasticity of the environment (please refer to our theoretical analysis in Appendix A.1).

### A.3 PRACTICAL EFFECTIVENESS OF ENTROPY MAXIMIZATION

We present in Table 5 the average error (penalty) predicted by EMO trained on 3 different datasets over: (1) Samples drawn from offline dataset $\mathcal{D}$; (2) Generated rollouts of horizon $H = 2$ using a random exploration policy; and (3) Generated rollouts of horizon $H = 5$ using a random exploration policy. Please note that the predicted penalty for each transition directly corresponds to the upper bound of total uncertainty attributed to the transition by EMO. The difference between the average predicted errors goes to show that our method of entropy maximization is indeed effective and trustworthy, as there is a distinguishable difference between uncertainty estimation over in-distribution data (samples from $\mathcal{D}$), and datasets that are dominated by OOD samples (generated rollouts). The difference between average predicted errors over rollouts of different horizons, however, depends on the value of hyperparameter $\alpha$. When $\alpha$ is large, the algorithm is more conservative, thus it is expected to see a small margin between the average error for rollouts of different horizons, as is the case for walker2d-medium and walker2d-medium-expert; but when $\alpha$ is comparatively smaller, we expect to see a larger margin between the average error over rollouts of varying horizons, as is the case of walker2d-medium-replay.

The difference between the relative scale of predicted errors over different datasets comes from the upper bound on the predicted entropy. We view the optimal relative scale of OOD uncertainty against in-distribution uncertainty as a function of environment and dataset charachteristics (e.g. coverage, optimality). As a result, in our practical implementation of EMO, we set an upper bound on the uncertainty prediction of the model, i.e. $\Sigma_\phi(s,a) \leq \Sigma_{max}, \forall (s,a) \in \mathcal{S} \times \mathcal{A}$, which indirectly controls the scale of OOD uncertainty against the in-distribution uncertainty. For each environment and dataset configuration, we treat $\Sigma_{max}$ as a hyperparameter, and optimize it along other hyperparameters of EMO. Although our observations are not conclusive and generalizable, we have observed that environment-dataset configurations which allow models to have better generalization (e.g. stable environments such as halfcheetah, and datasets that provide broad coverage such as medium datasets), will have smaller optimal values for $\Sigma_{max}$. In addition, we have also observed that optimality of the dataset can affect the optimal value of $\Sigma_{max}$. Datasets with more optimal transitions tend to have larger optimal values for $\Sigma_{max}$.

On top of that, we conduct another experiment to compare the average of predicted penalties of EMO and MOPO associated to OOD samples. In this experiment, we train the models of EMO and MOPO on walker2d-medium-replay dataset, and calculate the average of predicted penalties of each model on generated rollouts under the learned models using a random exploration policy. For rollout horizon $H = 2$, the average of predicted penalties for EMO was 7.198 compared to 2.818 of MOPO; and, for rollout horizon $H = 5$, the average of predicted penalties for EMO was 8.544 against 3.011 of MOPO, which shows that EMO takes a more conservative approach by penalizing an upper bound of uncertainty rather than penalizing a (potentially inaccurate) estimation of the uncertainty.

| Environment | Dataset | $\mathcal{D}$ | $H = 2$ | $H = 5$ |
|---|---|---|---|---|
| walker2d | medium | 1.340 | 11.386 | 11.465 |
| walker2d | medium-replay | 2.694 | 7.198 | 8.544 |
| walker2d | medium-expert | 1.273 | 30.962 | 31.177 |

Table 5: Average error predicted by EMO for 3 different datasets over: (1) Samples drawn from offline dataset $\mathcal{D}$; (2) Generated rollouts of horizon $H = 2$ using a random exploration policy; and (3) Generated rollouts of horizon $H = 5$ using a random exploration policy.

## A.4 COMPUTATION AND MEMORY EFFICIENCY

Although we cannot directly compare EMO against previous ensemble-based methods in terms of computation resource (as their implementations are different, which can heavily affect such measures), we still provide some indirect indications of the computation and memory efficiency of EMO. EMO manages to achieve this level of performance with about 0.34M parameters, while ensemble methods (MOPO, COMBO, RAMBO-RL) operate with around 1.1M parameters, which is an indirect indication that EMO improves upon SOTA in terms of memory and computational efficiency. In addition, we present in Figure 3 the performance progress of EMO against MOPO on walker2d-medium-expert dataset, averaged over 3 random seeds, where we can attribute the faster convergence rate of EMO as an indirect indicator of its computational efficiency compared to MOPO.

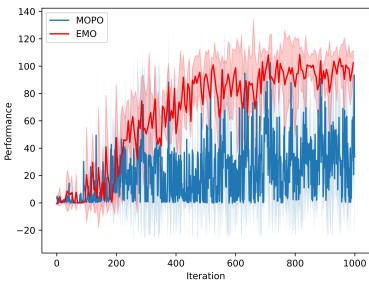

(a) Without moving average.

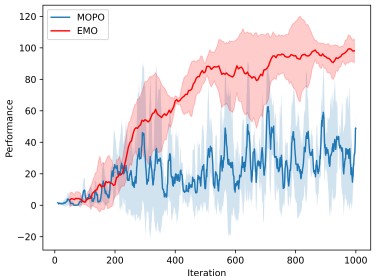

(b) Moving average of window size 10 for both EMO and MOPO.

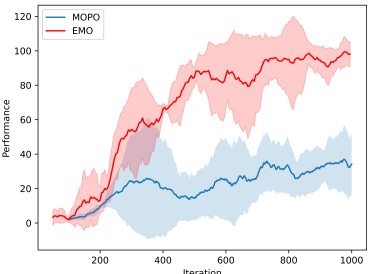

(c) Moving average of window size 10 for EMO and of size 100 for MOPO.

Figure 3: Performance progress of EMO against MOPO on walker2d-medium-expert dataset, averaged over 3 random seeds. The raw data is depicted in Figure 3a. In order to make the the raw figure more interpretable, we apply two configurations of moving averages over the results of EMO and MOPO.

A.5    REFERENCED ALGORITHMS

---

**Algorithm 4** Modified Version of EMO

---

**Require:** Offline data $\mathcal{D}$, batch size $b$, rollout horizon $h$, penalty coefficient $\lambda$, regularization coefficient $\alpha$
1: Initialize $\theta$ and $\phi$
2: Initialize policy $\pi$
3: **for** $K_1$ iterations **do**                                                                     ▷ Warm-up phase
4:        Sample a batch of transitions $\mathbb{B}_{\mathcal{D}}$ from the offline dataset $\mathcal{D}$.
5:        Compute $\mathcal{L}_1(\theta, \phi; \mathbb{B}_{\mathcal{D}})$ (Equation 2.)
6:        Compute gradients and update $\theta$ and $\phi$.
7: **end for**
8: **for** $K_2$ iterations **do**
9:        **for** $K_3$ iterations **do**
10:          Sample a batch of transitions $\mathbb{B}_{\mathcal{D}}$ from the offline dataset $\mathcal{D}$.
11:          Compute $\mathcal{L}_1(\theta, \phi; \mathbb{B}_{\mathcal{D}})$ (Equation 2.)
12:          Generate a batch of transitions $\bar{\mathbb{B}}_\pi$ using model rollouts (Algorithm 1.)
13:          Compute $\mathcal{L}_2(\phi; \bar{\mathbb{B}}_{\pi_e})$ (Equation 3.)
14:          Compute $\mathcal{L}(\theta, \phi; \mathbb{B}_{\mathcal{D}}, \bar{\mathbb{B}}_{\pi_e}) = \mathcal{L}_1(\theta, \phi; \mathbb{B}_{\mathcal{D}}) + \alpha \mathcal{L}_2(\phi; \bar{\mathbb{B}}_{\pi_e})$.
15:          Compute gradients and update $\theta$ and $\phi$.
16:        **end for**
17:        Define empirical MDP $\hat{M}$ with dynamics $\mu_\theta$ and reward $\tilde{r}(s, a) = r(s, a) - \lambda u(s, a)$, where
            $u(s, a) = \sqrt{\det \left( \Sigma_\phi(s, a) \right)}$.
18:        **for** $K_4$ iterations **do**
19:          Update $\pi$ on $\hat{M}$ with any arbitrary RL algorithm.
20:        **end for**
21: **end for**
22: **return** $\pi$

---

**Algorithm 5** Policy Optimization Method for Experiments

---

**Require:** pessimistic MDP $\hat{M} = (\mathcal{S}, \mathcal{A}, T_\mu, \tilde{r}, \gamma, \rho_0)$, rollout batch size $b$, rollout horizon $h$, offline dataset $\mathcal{D}$
1: Initialize policy $\pi$ and empty replay buffer $\mathcal{D}_{model} \leftarrow \emptyset$
2: **for** $K$ iterations **do**
3:        **for** $1, 2, ..., b$ (in parallel) **do**
4:          Sample the initial state $s_1$ of the rollout by sampling from offline data $\mathcal{D}$.
5:          **for** $j = 1, 2, ..., h$ **do**
6:              Sample an action based on the current policy of the agent $a_j \sim \pi(s_j)$.
7:              Compute the next state in the pessimistic MDP $s'_j = T_\mu(s_j, a_j)$.
8:              Compute the reward in the pessimistic MDP $\tilde{r}_j = \tilde{r}(s_j, a_j)$.
9:              Add transition $(s_j, a_j, \tilde{r}_j, s'_j)$ to $\mathcal{D}_{model}$.
10:          **end for**
11:        **end for**
12:        Draw samples from $\mathcal{D} \cup \mathcal{D}_{model}$, use SAC to update $\pi$.
13: **end for**

---

