# OpenReview forum: "Entropy-Regularized Model-Based Offline Reinforcement Learning"
_ICLR.cc/2023/Conference — Submitted to ICLR 2023_

### Official Review · Reviewer_hpPE · 2022-10-14

**Confidence:** 4
**Clarity, Quality, Novelty And Reproducibility:** All fine.
**Correctness:** 4
**Technical Novelty And Significance:** 3
**Empirical Novelty And Significance:** 2
**Recommendation:** 8

**Strength And Weaknesses:**

**Strengths**
- The paper addresses an important topic.
- The method delivers good results on the benchmarks studied.

**Weaknesses**

In my opinion, there is no major weakness that would justify rejection.

**Suggestions for improvement that should be implemented**
- The text still needs minor corrections. For example, lowercase is used several times where uppercase is necessary (e.g. "algorithm 5", "appendix A.2", "table 3", "equation 4"), while uppercase is used in at least one place where lowercase is required ("In this Algorithm,"). There are still broken sentences, like "that can lead a decrease in performance as well". In the bibliography, some words are wrongly written in lower case, like "fisher" or ""q-learning".

- When discussing uncertainty estimation without ensemble for offline RL, the older paper [1] should be discussed.

- Three repititions of the experiments are very few. The results could be significantly different in the long term mean. A standard deviation calculated using only three values is very unreliable.


**Suggestions for improvement in the future**
- The method should be tested on a stochastic benchmark.
When estimating reward uncertainty based on the modelling error, as in this paper, it is not ensured that only epistemic uncertainty is used and not aleatoric uncertainty. In case of deterministic environments, as used in this paper, this is not a problem because the aleatoric uncertainty is zero. In stochastic environments, however, it could be that the approach is sub-optimal because it also penalizes areatoric uncertainty, which is unnecessary and detrimental to the expected value of the performance.

**Further remark**

A +/- should always be followed by the uncertainty of the measured value. This is usually the standard error rather than the standard deviation in the case where the measured value is a mean. Ultimately, however, it is up to the experimenter (i.e., the authors in this case) to estimate the uncertainty appropriately. Thus, if the authors choose to report the standard deviation (which here is sqrt(3-1) times larger than the standard error) as the uncertainty, then the authors are choosing to make a conservative estimate of the uncertainty, which is perfectly fine, especially given the very small number of only three repititions.




[1] Depeweg et al., Decomposition of Uncertainty in Bayesian Deep Learning for Efficient and Risk-sensitive Learning, 2018

**Summary Of The Paper:**

The paper presents a new method for model-based offline RL.\
It uses both a simple uncertainty estimate based on the modeling error instead of an ensemble-based uncertainty estimate of the model, and maximizes the entropy of the model outside the data distribution. \
The method is tested on multiple datasets from two different deterministic MuJoCo benchmarks against various model-based and model-free offline RL methods and achieves competitive results.

**Summary Of The Review:**

A solid paper presenting another approach to avoid misestimation of the return in OOD situations in model-based offline RL.\
The proposed method is simple and effective according to the experiments. \
A few small improvements should be made. \
For the future, it would be important to investigate whether the approach is also effective in stochastic environments.

---

> ### Author Response · Authors · 2022-11-18
> **Response to reviewer hpPE**
>
> We sincerely thank the reviewer for the valuable comments and time dedicated to evaluating our work.
>
> **Comment 1:** The text still needs minor corrections.
>
> **Response 1:** Thank you very much for pointing out these mistakes. We have corrected them in the revised version of the draft.
>
> **Comment 2:** When discussing uncertainty estimation without ensemble for offline RL, the older paper [1] should be discussed.
>
> **Response 2:** Thank you very much for bringing this related work to our attention. Ensemble uncertainty quantification is technically a special case of uncertainty quantification in Bayesian neural networks with latent variables using nearest-neighbor methods, where each model in the ensemble corresponds to a sample from the posterior distribution over the parameters. We have included this discussion in our revised version of the paper (Introduction, paragraph 3).
>
> **Comment 3:** Three repititions of the experiments are very few.
>
> **Response 3:**     We would like to mention that our choice for the number of repetitions complies with prior work. MOReL reports the average results over 3 random seeds, CQL over 4 random seeds, and UWAC over 5 random seeds. For MOPO and COMBO, the results are reported over 6 random seeds.
>
> **Comment 4:** As a suggestion for future improvement, the method should be tested on a stochastic benchmark.
>
> **Response 4:** We do not explicitly test our method on stochastic environments, mainly because there are no widely used datasets and baselines for stochastic environments, since the application to stochastic environments is not extensively investigated in prior work. However, experimenting with stochastic environments would be an interesting venue for future work. Nevertheless, we have discussed this in detail in *"Applicability to stochastic environments"*, which you can refer to for more elaboration.
>
> **Comment 5:** Reporting standard error of the result might be a better option than reporting the standard deviation.
>
> **Response 5:**     Thank you very much for the insightful remark. Although we agree that in the case of sample-based measurements, standard error might be a better option, we chose to report standard deviation in order to comply with the results reported in the prior work, which also chose to report standard deviation instead of standard error (MOPO, COMBO, RAMBO-RL, etc.).
>
> [1] Depeweg et al., Decomposition of uncertainty in bayesian deep learning for efficient and risk-sensitive learning, ICML 2018.

---

> > ### Comment · Reviewer_hpPE · 2022-11-19
> > **Let's not make the shortcomings of others the standard**
> >
> >  Using only three values to calculate the mean leads to high uncertainties. If it is possible, more repetitions should be used to reduce the uncertainty of the results. If this is not possible, justification must be given as to why it was not possible to use more repetitions.
> >
> > The established practice in natural science publications is that each measurement result is reported with its uncertainty. The $\pm$ sign is used exclusively to indicate the uncertainty. The standard deviation is not a consistent measure for calculating the uncertainty, because while the uncertainty becomes smaller the more repetitions are made, the standard deviation does not.
> >
> > It is very regrettable that in some recent papers, these elementary rules were disregarded, this should by no means become common practice.

---

> > > ### Author Response · Authors · 2022-11-19
> > > **Response to reviewer HpPE (follow-up)**
> > >
> > > We agree that more repetitions results in a more accurate bound over the reported values. During the rebuttal period, and due to our time and computation constraints, we have only increased the number of repetitions for 3 of the walker2d datasets, as depicted in the table below. We will make sure to have the results over all the other datasets for the camera-ready version.
> > >
> > > | Environment| Dataset | EMO (3 seeds)| EMO (5 seeds)|
> > > |---|---|---|---|
> > > | walker2d | medium | $84.1 \pm 2.5$ | $82.6 \pm 1.7$ |
> > > | walker2d | medium-replay | $83.6 \pm 1.3$ | $82.5 \pm 3.1$ |
> > > | walker2d | medium-expert | $110.8 \pm 3.0$ | $106.4 \pm 5.0$ |
> > >
> > > We would also like to mention that in throughout the paper, we have changed our format for reporting the results from mean $\pm$ standard deviation to mean $\pm$ standard error.

---

### Official Review · Reviewer_EzoK · 2022-10-23

**Confidence:** 4
**Correctness:** 2
**Technical Novelty And Significance:** 2
**Empirical Novelty And Significance:** 2
**Recommendation:** 3

**Clarity, Quality, Novelty And Reproducibility:**

The paper is clearly written. The technical quality and novelty are limited. The authors provide their code in the supplementary material.

**Strength And Weaknesses:**

**Strengths**
- The paper studies an important topic which is uncertainty quantification for pessimistic offline RL. The paper is well-motivated as the ensemble methods do not offer reliable uncertainty estimation.
- The proposed approach has the benefit of decoupling pessimistic model-learning and policy-learning steps, similar to model-based methods MOReL and MOPO and unlike SoTA model-based offline RL methods COMBO and RAMBO-RL.
- The proposed algorithm is simple.
- In addition to evaluations on the D4RL benchmark, the authors also conduct ablation studies investigating the effect of exploration policy as well as regularization and regularization strength on learned policies.

**Weaknesses**
- The algorithm and technical contributions have limited novelty. The proposed algorithm is an instantiation of pessimistic lower confidence bound methods such as MOPO that subtract a penalty from rewards to account for partial coverage in offline data.
- It is unclear whether the algorithmic idea of building penalty terms based on determinants of covariance matrices is sound. It appears that the covariance is originally meant to capture the aleatoric uncertainty of the transitions (based on the definition on page 4) and later is used for the penalty terms, which instead require capturing epistemic uncertainty. No theoretical justification for this choice (even in a simple example) is presented.
- I feel the focus of the paper should be to justify why this particular uncertainty quantifier is sound and useful. So it's important to conduct some theoretical and/or empirical investigations showing that the approach actually gives better uncertainty quantification than other methods such as ensembles or the method in [1] instead of solely relying on the end-to-end result on offline RL benchmarks.
- The paper does not compare with model-based methods that go beyond using ensembles for uncertainty estimation such as the GELATO algorithm [1]. The paper also does not empirically compare with the ATAC algorithm [2]. Generally, the literature review is very incomplete.
- The empirical performance compared to existing methods is not very strong.
- The setting considered is limited. The algorithm focuses on the special case that transitions are Gaussian and uses learned covariance as an uncertainty quantification measure.
- Discussion in paragraph 2 in the Introduction regarding model-based vs. model-free methods in offline RL is not entirely accurate and the claims appear to be strong and not fully justified. For example, it is unclear whether model-free methods are *inherently* overly conservative or model-based methods are better at generalization. This claim is not supported in theoretical studies of offline RL and is also not entirely supported by empirical observations (for example, CQL (model-free) outperforms MOPO (model-based)).

**References**

[1] Guy Tennenholtz, Nir Baram, and Shie Mannor. Latent geodesics of model dynamics for offline reinforcement learning. In Deep RL Workshop NeurIPS 2021, 2021.

[2] Ching-An Cheng, Tengyang Xie, Nan Jiang, Alekh Agarwal Proceedings of the 39th International Conference on Machine Learning, PMLR 162:3852-3878, 2022.

**Summary Of The Paper:**

The paper studies pessimistic model-based offline reinforcement learning. Motivated by the fact that ensemble methods do not provide accurate uncertainty quantifiers used for constructing penalties subtracted from rewards, the authors present an entropy-regularized algorithm that learns a pessimistic model and provides uncertainty estimation outside the support of offline data. Empirical evaluations show comparable performance to SoTA offline RL algorithms on the D4RL benchmark.

**Summary Of The Review:**

It is unclear whether the approach presented in this paper is sound and useful. Theoretical and empirical justification is not presented for this choice of uncertainty quantifier. The paper misses empirical comparison with important related work. The approach has limited novelty and empirical performance is not very strong.

---

> ### Author Response · Authors · 2022-11-18
> **Response to reviewer EzoK**
>
> We sincerely thank the reviewer for the valuable comments and time dedicated to evaluating our work.
>
> **Comment 1:** The algorithm and technical contributions have limited novelty.
>
> **Response 1:** Although our method can be considered as an extension to MOPO, we would like to see its core value in: (1) eliminating the need for explicit uncertainty quantification by estimating an upper bound on the uncertainty instead; (2) using a single model of the environment and thus, being efficient in terms of memory and computation, while achieving competitive performance to SOTA; (3) being modular.
>
> **Comment 2:** It is unclear whether the algorithmic idea of building penalty terms based on determinants of covariance matrices is sound.
>
> **Response 2:**     As discussed in our revised theoretical formulation (see Appendix A.1), the penalty term can be any positive, increasing function of the entropy, which, in the case of Gaussian models, can be defined as the determinant of the covariance matrix. In our theoretical analysis, we show that EMO guarantees conservative policy evaluation and safe policy improvement regardless of the stochasticity of the environment, which goes to show that our proposed method is theoretically sound.
>
> Our method does not explicitly differentiate between aleatoric and epistemic uncertainties over OOD domains, and penalizes an upper bound on total uncertainty rather than epistemic uncertainty. We argue that it is reasonable to do so, as the aleatoric uncertainty itself is a source of error over OOD domains, since it cannot be reliably estimated over such domains. For a more detailed discussion about this, please refer to *"Applicability to stochastic environments"*.
>
> **Comment 3:** It is important to conduct investigations showing that EMO gives better uncertainty quantification than other methods.
>
> **Response 3:** We would like to mention that our method does not estimate the uncertainty itself, but estimates an upper bound on the uncertainty. As a result, our method might not provide a more accurate uncertainty quantification compared to methods like MOPO, but ensures that the uncertainty is not underestimated, by penalizing an upper bound on the uncertainty, rather than an estimation of the uncertainty itself, meaning that our algorithm will behave more conservatively over OOD samples than approaches which are based on quantifying the uncertainty.  To investigate this, we conduct an experiment to compare the average of predicted penalties of EMO and MOPO associated to OOD samples. In this experiment, we train the models of EMO and MOPO on walker2d-medium-replay dataset, and calculate the average of predicted penalties of each model on generated rollouts under the learned models using a random exploration policy. For rollout horizon $H=2$, the average of predicted penalties for EMO was $7.198$ compared to $2.818$ of MOPO; and, for rollout horizon $H=5$, the average of predicted penalties for EMO was $8.544$ against $3.011$ of MOPO, which shows that EMO takes a more conservative approach by penalizing an upper bound of uncertainty rather than penalizing a (potentially inaccurate) estimation of the uncertainty.
>
> **Comment 4:** Missing comparison with methods like GELATO and ATAC.
>
> **Response 4:** Thank you very much for bringing this related work to our attention. We have added ATAC as a model-free method based on adversarial training of actor and critic to the related work (Section 2). We have also added GELATO as a method that quantifies the uncertainty beyond bootstrap ensembles, by using  a  k-nearest neighbors approach, where the distance measure is defined as an approximate metric  on the learned (Riemannian) manifold in a latent space encoded by a VAE (Introduction, paragraph 4). The result of both methods have been included in the experiments section as well.
>
> **Comment 5:** Discussion about model-based vs. model-free methods is debatable and not fully justified.
>
> **Response 5:** Thank you very much for pointing out this matter. We agree that these claims might be debatable without comprehensive justification, which is not particularly in the scope of our work, and as a result, we have decided to remove these claims in the revised version of the draft.

---

### Official Review · Reviewer_bC5e · 2022-10-26

**Confidence:** 3
**Correctness:** 3
**Technical Novelty And Significance:** 2
**Empirical Novelty And Significance:** 2
**Recommendation:** 5

**Clarity, Quality, Novelty And Reproducibility:**

See the questions raised above. The reproducibility is justified by the code in supplements.

**Strength And Weaknesses:**

$\textbf{Strength:}$

This work attempt to estimate uncertainty in RL with a single model, which frees practitioners from deploying ensembles in uncertainty quantification. Estimating uncertainty with a single model is an important research direction in both offline RL and online exploration. The paper is well-written and easy to follow. The proposed EPO algorithm has better modularity than previous works, in the sense that it can incorporate various existing model-free RL algorithms (on top of its pessimistic environment estimation).


$\textbf{Weaknesses:}$

I have a few questions after reading the paper. Having some of those questions unresolved could be the potential weakness of the paper. Meanwhile, having some of those questions resolved could increase the potential impact of this paper.

$\textbf{Questions:}$

The claimed general approach seems to be limited by various model assumptions. In particular,

$\textbf{Q1:}$
How does EMO differentiate between aleatoric uncertainty and epistemic uncertainty? Gaussian modeling of next-step transition should also incorporate aleatoric uncertainty of $s'$ and $r$ unless the transition and reward are assumed to be deterministic (which is indeed the case for the MuJoCo environment in experiments though). Will such an assumption hinder the generality of the proposed EMO framework?


$\textbf{Q2:}$
The adopted (diagonal) Gaussian model requires continuous states, which may further hinder the generality of EMO.

$\textbf{Q3:}$
(Minor) In addition, the Gaussian model has only a single peak, which may limit the modeling of uncertainty in the next-step transition (there could only be one high-probability outcome).


In addition, I have a few question regarding uncertainty modeling in EPO.

$\textbf{Q4:}$
How does the explorative policy affect the performance of EPO? Is it trustworthy to consider data sampled from the explorative policy to be OOD data (the trajectory may overlap with the in-distribution dataset)? Why not using the modified EPO described in section 4.3, which seems to penalize more relevant OOD data? As a side remark, previous work (e.g., [1-4]) also sample OOD actions of training policy to enforce more relevant OOD penalization.

$\textbf{Q5:}$
The uncertainty estimate is maximized over OOD data in the training stage (the regularization phase). How does such maximization of OOD uncertainty affect the uncertainty estimate, in particular, the relative scale of OOD uncertainty against the in-distribution uncertainty? Is the estimated uncertainty (after maximization on OOD data) trustworthy?

Minor clarification question:


$\textbf{Q6:}$
How does EPO compare against previous ensemble-based methods in terms of the computation resource? I imagine that this could be one of the major advantages of utilizing a single model for uncertainty estimation.


[1] A. Kumar et al., Conservative Q-Learning for Offline Reinforcement Learning. (2020)

[2] I. Kostrikov et al., Offline Reinforcement Learning with Fisher Divergence Critic Regularization. (2021)

[3] C. Bai et al., Pessimistic Bootstrapping for Uncertainty-Driven Offline Reinforcement Learning. (2022)

[4] Lyu et al., Mildly Conservative Q-Learning for Offline Reinforcement Learning. (2022)

**Summary Of The Paper:**

This paper presents EPO, a novel model-based offline RL algorithm. The highlight of EPO is that it utilizes only a single model to estimate uncertainty, in contrast with previous works that require an ensemble of models.

The algorithm has multiple phases. In phase one, the algorithm fits a Gaussian model for the transition and reward. In phase two the algorithm maximizes the differential entropy of the fitted Gaussian model over an exploratory dataset, which is collected by some exploratory policy on the fitted environment from phase one. Finally, EPO optimizes the model over the fitted environment with an uncertainty penalization (based on the entropy of the fitted model) over the reward.

The authors conducted experiments on the MuJoCo environment and showed that EPO is competitive with some offline RL SOTAs that utilize an ensemble of models for uncertainty quantification.

**Summary Of The Review:**

The authors present an interesting attempt at uncertainty quantification with a single model. The approach appears to be promising at least in the test environments. Nevertheless, the underlying model assumptions with the Gaussian distribution could be strong, which may limit the generality of EPO as a general workflow.

---

> ### Author Response · Authors · 2022-11-18
> **Response to reviewer bC5e**
>
> We sincerely thank the reviewer for the valuable comments and time dedicated to evaluating our work.
>
> **Comment 1:** The adopted (diagonal) Gaussian model requires continuous states, which may further hinder the generality of EMO.
>
> **Response 1:** We would like to mention that in this paper, we only focus on continuous environments, which complies with the relevant prior work in model-based offline RL (MOPO, MOReL, COMBO, RAMBO-RL, etc.), as learning a model for discrete environments corresponds to a different literature, that is outside the scope of this work.
>
> **Comment 2:** Is a random exploration policy trustworthy? Why not prefer modified EMO?
>
> **Response 2:** As long as $\pi_{e}(a | s) \neq \pi^{B}(a | s)$, we are guaranteed to generate OOD data by generating rollouts based on $\pi_{e}$. However, we cannot guarantee that the generated samples are exclusive to OOD domains. Depending on $\pi_{e}(a | s)$ and how much it differs from $\pi^{B}(a | s)$, the proportion of OOD samples to in-distribution samples might change. Although the generated rollouts might overlap with the in-distribution dataset, the effect of entropy maximization over the generated rollouts is dominated by negative log-likelihood when such overlaps happen, as we assume the regularization coefficient $\alpha$ in our formulation of loss function to be a small value. Thus, entropy maximization only occurs over OOD domains. However, as you have stated in your question as well, it is important to ensure that generated OOD samples under $\pi_{e}$ are relevant. If we maximize the entropy over OOD domains that will not be explored throughout policy optimization phase, then our method will not perform as expected. The best way to produce relevant OOD data is by utilizing the current policy of the agent (as discussed in the modified version of EMO), as entropy maximization is applied to OOD transitions that are more likely to be explored by the agent. However, modified EMO is task specific and computationally inefficient, and it constantly changes the model throughout policy optimization. Since we are motivated by modularity and computational efficiency, we prioritize uniform distribution as our exploration policy, since it prevents active alteration of the model during policy optimization, and allows for a modular framework. Note that since the coverage of rollouts generated under a uniform exploration policy subsumes the coverage of rollouts under the current policy of agent, relevant OOD transitions will still be included in the entropy maximization process, which makes our framework trustworthy, but we cannot deny that using the current policy of the agent results in more relevant OOD data, and subsequently, better performance.

---

> > ### Author Response · Authors · 2022-11-18
> > **Response to reviewer bC5e (follow-up)**
> >
> > **Comment 3:** How does maximization of OOD uncertainty affect the uncertainty estimate?
> >
> > **Response 3:** In the following table, we present the average error (penalty) predicted by EMO trained on 3 different datasets over: (1) Samples drawn from offline dataset $\mathcal{D}$; (2) Generated rollouts of horizon $H=2$ using a random exploration policy; and (3) Generated rollouts of horizon $H=5$ using a random exploration policy. Please note that the predicted penalty for each transition directly corresponds to the upper bound of total uncertainty attributed to the transition by EMO. The difference between the average predicted errors goes to show that our method of entropy maximization is indeed effective and trustworthy, as there is a distinguishable difference between uncertainty estimation over in-distribution data (samples from $\mathcal{D}$), and datasets that are dominated by OOD samples (generated rollouts). The difference between average predicted errors over rollouts of different horizons, however, depends on the value of hyperparameter $\alpha$. When $\alpha$ is large, the algorithm is more conservative, thus it is expected to see a small margin between the average error for rollouts of different horizons, as is the case for walker2d-medium and walker2d-medium-expert; but when $\alpha$ is comparatively smaller, we expect to see a larger margin between the average error over rollouts of varying horizons, as is the case of walker2d-medium-replay.
> >
> > | Environment | Dataset  | $\mathcal{D}$  | $H=2$  | $H=5$  |
> > |---|---|---|---|---|
> > | walker2d  | medium  |  $1.340$ | $11.386$  | $11.465$  |
> > | walker2d  | medium-replay  | $2.694$  | $7.198$  | $8.544$  |
> > | walker2d  | medium-expert  | $1.273$  | $30.962$  | $31.177$  |
> >
> > The difference between the relative scale of predicted errors over different datasets comes from the upper bound on the predicted entropy. We view the optimal relative scale of OOD uncertainty against in-distribution uncertainty as a function of environment and dataset charachteristics (e.g. coverage, optimality). As a result, in our practical implementation of EMO, we set an upper bound on the uncertainty prediction of the model, i.e. $\Sigma_{\phi}(s, a) \leq \Sigma_{max}, \forall (s, a) \in \mathcal{S}\times\mathcal{A}$, which indirectly controls the scale of OOD uncertainty against the in-distribution uncertainty. For each environment and dataset configuration, we treat $\Sigma_{max}$ as a hyperparameter, and optimize it along other hyperparameters of EMO. Although our observations are not conclusive and generalizable, we have observed that environment-dataset configurations which allow models to have better generalization (e.g. stable environments such as halfcheetah, and datasets that provide broad coverage such as medium datasets), will have smaller optimal values for $\Sigma_{max}$. In addition, we have also observed that optimality of the dataset can affect the optimal value of $\Sigma_{max}$. Datasets with more optimal transitions tend to have larger optimal values for $\Sigma_{max}$. We have also included this discussion in Appendix A.3 in the revised draft.
> >
> > **Comment 4:** How does EMO compare against previous ensemble-based methods in terms of the computation resource?
> >
> > **Response 4:** Although we cannot directly compare EMO against previous ensemble-based methods in terms of computation resource (as their implementations are different, which can heavily affect such measures), we still provide some indirect indications of the computation and memory efficiency of EMO. EMO manages to achieve this level of performance with about 0.34M parameters, while ensemble methods (MOPO, COMBO, RAMBO-RL) operate with around 1.1M parameters, which is an indirect indication that EMO improves upon SOTA in terms of memory and computational efficiency. In addition, we present in Figure 3 in Appendix A.4 the performance progress of EMO against MOPO on walker2d-medium-expert dataset, averaged over 3 random seeds, where we can attribute the faster convergence rate of EMO as an indirect indicator of its computational efficiency compared to MOPO. We have also included this discussion in Appendix A.4 in the revised draft.

---

### Official Review · Reviewer_Y21Y · 2022-10-30

**Confidence:** 3
**Correctness:** 3
**Technical Novelty And Significance:** 1
**Empirical Novelty And Significance:** 2
**Recommendation:** 5

**Clarity, Quality, Novelty And Reproducibility:**

To me, the novelty seems light. The Gaussian structure of the model parameters makes me expect some theoretical guarantees, and perhaps also, a theoretical insight on how to choose the regularization coefficient depending on the size of the dataset and/or its support.

Questions to the authors:
1. In sec. 4.2 when reporting the overall performance, the authors state that « the values for RAMBO-RL, etc., are reported from their respective papers »: does this yield a fair comparison? I would expect re-running those baselines and comparing results on the same machine+seeds.
2. The authors state that avoiding ensemble model using regularization instead increases sample complexity. I do not find any empirical evidence of this claim in the experiments. Therefore, although intuitive, such a claim is still arguable (no theoretical guarantees plus no empirical evidence)

Minor comments:
- p.2, §2: ‘utlizing’ -> ‘utilizing’
- p.2, §2: ‘the best of two worlds’ -> ‘the best of both worlds’
- p.7, §1: ‘adheres to the prior work’ -> ‘adheres to prior work’
- p. 7 §1: ‘to train an RL algorithm to obtain $\pi^*$’ —> redundant ‘to’
- section 3.4.: I would merge this section to the previous one (too short)
- p.7, sec. 4,2: ‘to the state-of-the-art’ —> ‘to state-of-the-art’

**Strength And Weaknesses:**

The paper is clearly written and easy to follow. However, although I am not familiar with the offline RL literature, I found the proposed method quite standard, ie, regularizing the loss for better generalization.
Also, I am a bit skeptical regarding the experimental results. I explain why below.


**Summary Of The Paper:**

This work analyses model-based approaches to offline RL. Instead of considering model ensemble, it suggests using entropy regularization as a way to explore beyond the data support.

**Summary Of The Review:**

The lack of technical novelty and my concerns regarding experiments lead me to a weak accept.

---

> ### Author Response · Authors · 2022-11-18
> **Response to reviewer Y21Y**
>
> We sincerely thank the reviewer for the valuable comments and time dedicated to evaluating our work.
>
> **Comment 1:** Reporting the results of other methods from their respective papers might not result in a fair comparison.
>
> **Response 1:** As was done by almost all the prior work (e.g. MOPO, MOReL, COMBO, RAMBO-RL, etc.), it is a common practice to directly refer to the results of other papers; thus, our choice for directly borrowing the results from other papers complies with prior work in that sense.
>
> **Comment 2:** The authors state that avoiding ensemble models using regularization increases sample efficiency, which is not supported by any evidence.
>
> **Response 2:** We would like to apologize for being unclear, but we do not have any evidence that training via regularization improves sample efficiency any more than other model-based methods (e.g. ensemble methods), and we did not intend to indicate that. In the paper, whenever we are referring to an improvement in sample efficiency, we are referring to improvement in model-based methods compared to model-free methods, not improvement in training via regularization compared to other model-based methods. We have clarified this in our revised version of the draft as well.
>
> **Comment 3:** Minor comments (typos and structure).
>
> **Response 3:** Thank you very much for pointing out these mistakes. We have corrected them in the revised version of the draft, except for merging Section 3.4 with Section 3.3; although we agree that Section 3.4 is short, we believe these two sections are not related enough to be merged with one another.

---

> ### Author Response · Authors · 2022-11-18
> **Mismatch of the scores**
>
> We would like to mention that there is a mismatch between your evaluation in the summary of the review (weak accept) and the recommendation (marginal reject). Respectfully, we would like to ask you if it is possible to clarify upon this mismatch.

---

### Official Review · Reviewer_8Cwi · 2022-10-31

**Confidence:** 4
**Correctness:** 3
**Technical Novelty And Significance:** 2
**Empirical Novelty And Significance:** 2
**Recommendation:** 5

**Clarity, Quality, Novelty And Reproducibility:**

The method itself is easy to understand and looks clearly written. Still, I am not convinced of why we should maximize the entropy: Due to its nature of directly using entropy as a reward penalty, it seems that the method is hard to be applied to stochastic environments.

**Strength And Weaknesses:**

[Strengths]
1. This paper presents a method that provides uncertainty quantification using a single model. This is favorable in terms of computational efficiency, compared with the existing model-based offline RL methods that use ensembles.


[Weaknesses]
1. It seems that the method cannot be applied to stochastic environments. In the proposed framework, it is unclear how to differentiate between aleatoric uncertainty and epistemic uncertainty when using a learned covariance matrix for uncertainty quantification. I think we may need to penalize epistemic uncertainty but not penalize aleatoric uncertainty.
2. Mixed experimental results: EMO outperforms baselines in some domains but underperforms in other domains.
3. Modified EMO shares a similar spirit with RAMBO-RL, in terms of adversarial model update, but RAMBO-RL still outperforms Modified EMO in Table 3.
4. In Figure 2, it seems the performance of EMO is highly sensitive to the hyperparameter $\alpha$.

[Questions]
1. Do $\mu_\theta$ and $\Sigma_\phi$ share the lower-layer network parameters? or are they completely separate networks?


**Summary Of The Paper:**

This paper presents Entropy-regularized Model-based Offline RL (EMO), a model-based offline RL algorithm that learns a pessimistic MDP, where the uncertainty quantification is performed without using ensemble models. The dynamics model is trained in two phases. In the first warm-up phase, the model is learned by maximizing the log-likelihood. In the second regularization phase, an additional dataset is generated by an exploration policy on the learned model, and the model is optimized by maximizing the log-likelihood for the offline dataset while maximizing the entropy for the generated dataset by the exploration policy. In the experiments, EMO performs competitively with the baseline algorithms.


**Summary Of The Review:**

Although the proposed method is appealing in that it learns only a single model for uncertainty estimation, the experimental results are not convincing enough by mixed results. Also, I have a concern that the method would be difficult to deal with stochastic environments since aleatoric uncertainty and epistemic uncertainty cannot be distinguished in the proposed framework.

---

> ### Author Response · Authors · 2022-11-18
> **Response to reviewer 8Cwi**
>
> We sincerely thank the reviewer for the valuable comments and time dedicated to evaluating our work.
>
> **Comment 1:** In Figure 2, it seems the performance of EMO is highly sensitive to the hyperparameter $\alpha$.
>
> **Response 1:** We have updated Figure 2 in our revised version of the draft. In the updated figure, we have added extra datapoints and increased the number of repetitions from 3 random seeds to 5 for the original EMO runs to clarify our discussion about the effect of hyperparameter $\alpha$ on the performance of EMO. Please note that the values of $\alpha$ in this experiment were chosen to investigate how EMO behaves over different *ball parks* of this hyperparameter, and not to explore the sensitivity around a fine-tuned value, as $\alpha$ varies from 0.0001 to 2, and the horizontal axis in Figure 2 is not linear.
>
> **Comment 2:** Do $\mu_{\theta}$ and $\Sigma_{\phi}$ share the lower-layer network parameters? or are they completely separate networks?
>
> **Response 2:** $\mu_{\theta}$ and $\Sigma_{\phi}$ are two outputs of a single neural network, i.e. they share the same network, except for their output layers. We have clarified your question in our revised version of the draft (Section 4.1, first paragraph) as well.

---

### Author Response · Authors · 2022-11-18
**Gaussian structure of model**

This comment is in response to comments from reviewers Y21Y, bC5e, and EzoK, regarding the Gaussian structure of model in EMO, and whether it is a limiting factor in EMO.

**Response:**

As discussed in the revised version of our theoretical analysis in Appendix A.1, our theory is developed around a model with an arbitrary distribution. Thus, in practice, the model can be characterized by any arbitrary distribution, as there is no theoretical benefit from using a Gaussian distribution based on our analysis. In our practical implementation, our choice for characterizing the model with a Gaussian distribution is motivated by: (1) It complies with prior work. Almost all related model-based methods characterize their model of the environment with a Gaussian distribution (MOPO, MOReL, COMBO, RAMBO-RL); and (2) Gaussian distribution models the next state and reward with a mean and an estimated uncertainty around the predicted mean, which, intuitively, is a more reasonable choice in the context of offline RL with uncertainty estimation, as the uncertainty is separately estimated by the Gaussian distribution.

---

### Author Response · Authors · 2022-11-18
**Experimental results**

This comment is in response to comments from reviewers 8Cwi, Y21Y, and EzoK, regarding mixed experimental results, and EMO not being dominantly stronger than other methods in conducted experiments.

**Response:**

Our goal is not to outperform every method, but to propose an approach that can be used properly with the application/data at hand. One method might be suitable when the behavior policy is random, another one for more expert data, etc. In the same sense, when modularity and/or computation and memory efficiency is a priority, EMO will be a suitable choice.

We would like to see the core value of our work as a computationally efficient, modular framework that performs competitively to SOTA, instead of a method that outperforms other methods. Neither using a single model (which improves computation and memory efficiency) nor using a random exploration policy (which results in modularity) will result in an increase in performance, so our design choices are not inspired only by increasing the final performance. Even in the case of modified EMO, where we sacrifice the modularity of original EMO to use the current policy of the agent for model optimization (similar to RAMBO-RL), RAMBO-RL still has a very obvious advantage (as long as performance is concerned) over EMO in that it uses an ensemble of models instead of a single model that modified EMO has.

We would also like to mention that EMO manages to achieve this level of performance with about 0.34M parameters, while ensemble methods (MOPO, COMBO, RAMBO-RL) operate with around 1.1M parameters, which goes to show that EMO improves upon SOTA in terms of memory and computational efficiency.

---

### Author Response · Authors · 2022-11-18
**Applicability to stochastic environments**

This comment is in response to comments from reviewers 8Cwi, bC5e,  EzoK, and hpPE, regarding differentiation between aleatoric and epistemic uncertainties and applicability to stochastic environments.

**Response:**

Our model does not explicitly differentiate between aleatoric and epistemic uncertainties over OOD domains, and penalizes the rewards based on an upper bound over total uncertainty, instead of only epistemic uncertainty. However, we argue that this is a good approach in practice, even in the case of stochastic environments. In general, uncertainty, be it aleartoric or epistemic, is a source of error in RL algorithms. If we have a reliable and accurate estimation of either of these uncertainties, we can calculate their effect in our evaluations and algorithms. Otherwise, we should find ways to indirectly account for these sources of error, in order to prevent our methods from exploiting potential errors caused by these sources (e.g. by forming performance lower bounds with reward penalties etc.).

Same as any other measure, we argue that aleatoric uncertainty cannot be reliably quantified over OOD domains either, especially when we only have a single model; even if we have an ensemble of models, where we can estimate the aleatoric uncertainty by averaging over the variances of each model, the estimation would be inaccurate, as in practice, the sample size from the posterior distribution of parameters (the number of models in the ensemble) is small and samples are potentially correlated for the reasons discussed in the paper. Thus, aleatoric uncertainty over OOD domains becomes an unmeasurable source of error itself. As a result, it is reasonable to penalize a measure of total uncertainty over OOD domains, rather than only penalizing epistemic uncertainty, as we cannot confidently rely on the estimated aleatoric uncertainty over such domains. As for the in-distribution aleatoric uncertainty, we will show that our method preserves the reliable estimation of in-distribution aleatoric uncertainty. For this, we will discuss the characteristics of a model trained based on EMO over the support of data as well as OOD domains. Recall the formulation of the proposed hybrid loss in Equation (4), where $L_{1}$ corresponds to NLL loss, $L_{2}$ corresponds to entropy regularization term, and $\alpha$ is the regularization coefficient. Please note that as discussed in Section 3.2.2 of the paper, we assume $\alpha$ has a small value.

(i) Since we assume that $\alpha$ is small, hybrid loss will be dominated by the NLL loss ($L_{1}$) over the support of offline data. As a result, $\Sigma_{\phi}(s, a)$ will actually correspond to the aleatoric uncertainty over those regions (which, in the case of deterministic environments, is close to zero). Note that this estimation is reliable since it is learned over the support of offline data.

(ii) Over OOD domains, NLL loss ($L_{1}$) does not exist (as we do not have any supervised data over OOD domains), and hybrid loss will be dominated by entropy regularization term. Thus, $\Sigma_{\phi}(s, a)$ will correspond to an upper bound of total uncertainty/error over OOD domains.

As a result, the estimated aleatoric uncertainty over the in-distribution data, is preserved by $\Sigma_{\phi}(s, a)$, as discussed in (i). In the following table, we also present the average value of $\sqrt{\det(\Sigma_{\phi}(s, a))}$ over the samples drawn from the offline dataset for: (1) a model trained based on EMO; and (2) an NLL model, which goes to show that the uncertainty quantification is practically unaffected over in-distribution data, meaning that the reliable aleatoric uncertainty is preserved in a model trained by EMO, which can be accounted for with minimal modifications to the original EMO algorithm, e.g. by excluding it from the penalty term.

|  Environment | Dataset  | EMO  | NLL  |
|---|---|---|---|
| walker2d | medium  |  $1.340$ |  $1.414$ |
| walker2d |  medium-replay | $2.694$  | $2.792$  |
| walker2d | medium-expert  |  $1.273$ | $1.378$  |

On another note, we would also like to emphasize that EMO in its original form, theoretically guarantees conservative policy evaluation and safe policy improvement, regardless of the stochasticity of the environment (please refer to the revised version of our theoretical analysis in Appendix A.1).

---

> ### Comment · Reviewer_hpPE · 2022-11-30
> **Good point on practice**
>
> I agree that in practice it is rather the case that state-action pairs (s,a) leading to a large number of possible successor states s' are more likely to be unfavorable (compared to those with a small number).
>
> In addition, for the same number of visits N(s,a), a large number of possible successor states also leads to larger epistemic uncertainty.
>
> It is certainly possible to construct a benchmark where state-action pairs leading to a large number of possible successor states both promise a particularly high return and are visited so frequently that the epistemic uncertainty is not greater there than elsewhere---but of course this says nothing about practical environments.

---

### Author Response · Authors · 2022-11-18
**Summary of revisions**

Based on the reviewers' comments, we have clarified the arguments, structure, and theoretical descriptions and added some new experiments. Changes in the revised paper are shown in blue. We summarize the major changes below:

 1. **Added discussions** about additional related work in introduction and related work. [Reviewers EzoK, hpPE]
 2. **Rewritten the theoretical analysis** in Appendix A.1 and Section 3.4. [All reviewers]
 3. **Added comparisons** with GELATO and ATAC (Table 1 and Section 4.2). [Reviewer EzoK]
 4. **Updated figure** in Figure 2. [Reviewer 8Cwi]
 5. **Added discussions and experiments** about applicability to stochastic environments in Appendix A.2. [Reviewers 8Cwi, bC5e, EzoK, hpPE]
 6. **Added discussions and experiments** about practical effectiveness of entropy maximization in Appendix A.3. [Reviewer bC5e]
 7. **Added discussions and experiments** about computation and memory efficiency in Appendix A.4. [Reviewer bC5e]
 8.  **Changed format** of the reported results from mean $\pm$ standard deviation to mean $\pm$ standard error. [Reviewer hpPE]

We have also corrected the minor issues [Reviewer Y21Y, hpPE], and made minor revisions to the text for space considerations.

We would appreciate it if the reviewers can please take a look at these changes and let us know if they have any other concerns.

---

### Decision · Program_Chairs · 2023-01-20

**Decision:**

Reject

**Justification For Why Not Higher Score:**

While experimental results look great, additional experiments needs to be done (stochastic domains, sensitivity of hyperparameters, etc) and included in the paper.

**Justification For Why Not Lower Score:**

N/A

**Metareview: Summary, Strengths And Weaknesses:**

This paper presents a model-based offline RL approach. Assuming the diagonal Gaussian distribution for the transition and reward model, the main idea is to add a regularization term that increases the entropy on OOD samples generated by an exploration policy to model the uncertainty. This is a neat idea while most of the previous work adopts computation-heavy ensemble models to calculate the uncertainty of OOD samples.

Yet, reviewers raised concerns about not being able to differentiate between epistemic vs aleatory uncertainty. While the authors argue against these comments by emphasizing superb experimental results, more investigation is needed from experiments on stochastic domains. In fact, there are a lot of design choices involved (such as hyperparameters such as alpha, K1, K2 in EMO, choice of exploration policy) so with a significant tuning one can obtain pretty good results. The theoretical analysis in the appendix seems quite straightforward, and the abstract still emphasizes the method measures epistemic uncertainty (which isn’t).